# Distributions of Cisco (*Coregonus artedi*) in the upper Great Lakes in the mid-twentieth century, when populations were in decline

Yu-Chun Kao[1]*, Renee E. Renauer-Bova[1], David B. Bunnell[1], Owen T. Gorman[2], Randy L. Eshenroder[3]

1 U.S. Geological Survey Great Lakes Science Center, Ann Arbor, Michigan, United States of America, 2 U. S. Geological Survey Great Lakes Science Center, Lake Superior Biological Station, Ashland, Wisconsin, United States of America, 3 Great Lakes Fishery Commission, Ann Arbor, Michigan, United States of America

* ykao@usgs.gov

**Data Availability Statement:** All data used in this study can be found in a U.S. Geological Survey data release: doi.org/10.5066/P9P28C5V.

## Abstract

The restoration of the once abundant Cisco (*Coregonus artedi*) is a management interest across the Laurentian Great Lakes. To inform the restoration, we (1) described historical distributions of Cisco and (2) explored whether non-indigenous Rainbow Smelt (*Osmerus mordax*) and Alewife (*Alosa pseudoharengus*) played a role in the decline of Cisco populations across the upper Great Lakes (i.e., Lakes Superior, Michigan, and Huron). Our source data were collected from fishery-independent surveys conducted by the U.S. Fish and Wildlife Service's research vessel R/V *Cisco* in 1952–1962. By analyzing data collected by gill-net surveys, we confirmed the importance of embayment and shallow-water habitats to Cisco. We found that Cisco was abundant in Whitefish Bay and Keweenaw Bay, Lake Superior, and in Green Bay, Lake Michigan, but we also found a sign of Cisco extirpation in Saginaw Bay, Lake Huron. Our results also showed that Ciscoes generally stayed in waters <80 m in bottom depth throughout the year. However, a substantial number of Ciscoes stayed in very deep waters (>150 m in bottom depth) in summer and fall in Lake Michigan, although we cannot exclude the possibility that these Ciscoes had hybridized with the other *Coregonus* species. By comparing complementary data collected from bottom-trawl surveys, we concluded that the spatiotemporal overlap between Rainbow Smelt and Cisco likely occurred across the upper Great Lakes throughout 1952–1962. These data were consistent with the hypothesis that Rainbow Smelt played a role in the decline of Cisco populations across the upper Great Lakes in the period. We also found that the spatiotemporal overlap between Alewife and Cisco likely occurred only in Saginaw Bay in fall 1956 and in Lake Michigan after 1960. Thus, any potential recovery of Cisco after the 1950s could have been inhibited by Alewife in Lakes Michigan and Huron.

**Funding:** This study was supported by the U.S. Environmental Protection Agency's Great Lakes Restoration Initiative, Habitat and Species Focus Area, Coregonine Template for fiscal year 2018, in the form of a grant to YCK, RER, DBB, and OTG."

**Competing interests:** The authors have declared that no competing interests exist.

## Introduction

Understanding the causes of declining biodiversity or extirpation of species is critical to the success of conservation and restoration efforts, which typically seek to maintain or increase species richness and sustain key ecosystem services [1–3]. For example, part of the process of determining whether to provide protection to a species at risk in the United States and Canada is to conduct a formal assessment of threats [4]. The challenge of determining the underlying causes of declining biodiversity or distribution of species has been aggravated over time as most ecosystems have been exposed to multiple anthropogenic stressors that could be responsible [5, 6]. For the restoration of species extirpated long ago, this challenge could be even greater due to data limitations [7, 8] or ecosystem regime shifts [9]. In many cases, scientists have attempted to identify the cause of historical extirpation through a combination of analyzing data for historical distribution and reconstructing the chronology of critical stressors, such as overexploitation, interactions with non-indigenous species, cultural eutrophication, and, more recently, climate change [10–13].

In the upper Laurentian Great Lakes (i.e., Lakes Superior, Michigan, and Huron; Fig 1), managers have varying levels of interest in the conservation and restoration of a native fish—Cisco (*Coregonus artedi*) [14]. The word "cisco" (plural ciscoes) can be confusing as it has been also used to describe the non-whitefish fishes of coregonines (Subfamily Coregoninae) in the subgenus *Leucichthys* [15]. To distinguish the difference, we capitalized the common names of fishes in this paper, so that "Cisco(es)" is referring to the species *C. artedi*, while "ciscoes," always used in plural, is referring to a *Leucichthys* assemblage. Cisco is the only shallow-water species in the *Leucichthys* assemblage of the Great Lakes, which originally included another seven species collectively known as deepwater ciscoes—*C. alpenae*, *C. hoyi*, *C. johannae*, *C. kiyi*, *C. nigripinnis*, *C. reighardi*, and *C. zenithicus*. *C. artedi* is still present across the upper Great Lakes, although the population sizes are not comparable to historical levels [16]. All seven deepwater ciscoes were present in Lakes Michigan and Huron but only *C. hoyi* is still present today [16]. Five deepwater ciscoes were present in Lake Superior, of which three (*C. hoyi*, *C. kiyi*, and *C. zenithicus*) are still present and two (*C. nigripinnis* and *C. reighardi*) have uncertain status [16].

Historically, Ciscoes were important economically and ecologically due to their large populations across the upper Great Lakes. Cisco is a species with many morphological forms [15, 16, 18, 19]. In the early twentieth century, Koelz [15, 19] described three Cisco forms in the Great Lakes and more than 20 Cisco forms in the nearby waterbodies. The dominant Cisco form in the upper Great Lakes was described by Koelz as "typical *artedi*." Commercial landing records showed that Cisco, mostly the typical *artedi* form, was the most harvested species in terms of biomass in each of the upper Great Lakes between the late nineteenth century and the mid-twentieth century [20]. In the historical food webs, Ciscoes played an important role in the pelagic food chains, feeding mainly on zooplankton [15] and being preyed upon by predators such as Lake Trout (*Salvelinus namaycush*) and Burbot (*Lota lota*) [21]. Additionally, Ciscoes subsidized nearshore biological communities by migrating to shallow waters to spawn in late fall [22, 23]. However, the importance of Ciscoes started to decline with their declining population sizes in the mid-twentieth century. In Lake Superior, Cisco commercial landings have been at a low but consistent level since the late 1970s, about 15% of the historical peak in the 1950s [20]. In Lakes Michigan and Huron, the commercial fisheries of Cisco collapsed in the 1960s [20] but there are signs of Cisco recovering since the mid-2000s. Increases in Cisco populations have been reported in northern Lake Michigan [24] and in the North Channel and Georgian Bay of Lake Huron [25], although the relatedness of these contemporary Cisco to the historical forms of Cisco in these regions has been reassessed [16, 25, 26].

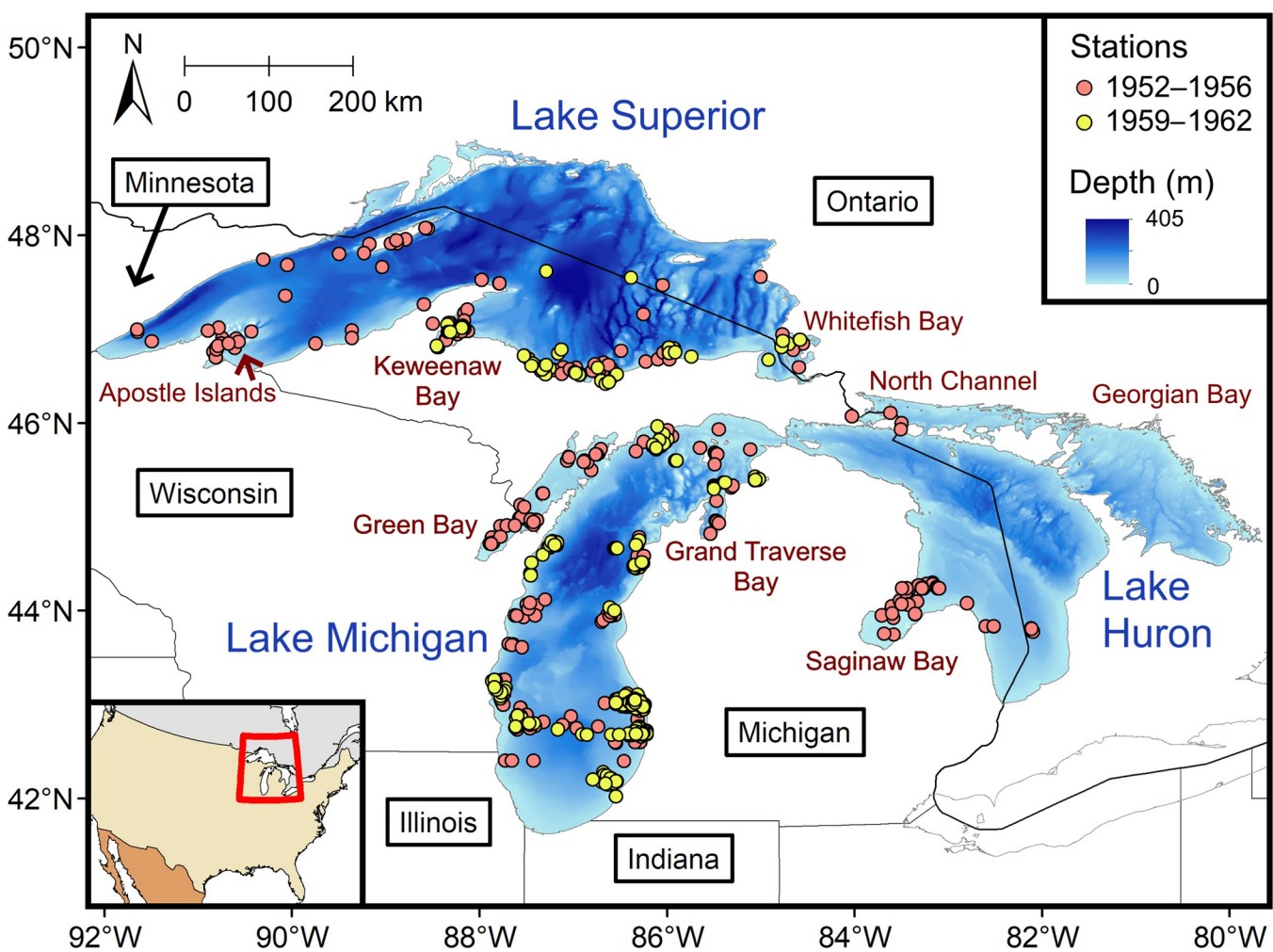

**Fig 1. Upper Great Lakes map.** Circles in different colors are the survey stations of the R/V *Cisco*, a research vessel of the U.S. Fish and Wildlife Service, in 1952–1956 and 1959–1962 in Lakes Superior, Michigan, and Huron. The map was generated with geographic information system (GIS) software ArcGIS version 10.8 (http://www.esri.com/software/arcgis/). The GIS layers for the survey stations were generated in this study. The GIS layers for the Great Lakes bathymetry and polygons were obtained from the Great Lakes Aquatic Habitat Framework's [17] public domain spatial database (https://glahf.org/data/). The GIS layers for political boundaries were obtained from another public domain spatial database Natural Earth (https://www.naturalearthdata.com).

The sharp decline of Cisco populations in the upper Great Lakes in the mid-twentieth century has been attributed to overfishing, habitat degradation, and interactions with non-indigenous species [27–30]. However, the relative importance of each of these stressors is still unclear. Overfishing was an apparent cause of the initial decline of Cisco populations between the 1890s and 1920s [30], but the populations could still support sizeable commercial landings until the 1950s [20]. While severe habitat degradations driven by eutrophication had occurred in areas such as Green Bay of Lake Michigan [31] and Saginaw Bay of Lake Huron [32], the upper Great Lakes were mostly oligotrophic and mesotrophic in the mid-twentieth century and likely less prone to egg mortality owing to low dissolved oxygen concentrations [33–35]. The decline of Cisco populations has also been linked to the proliferation of non-indigenous Alewife (*Alosa pseudoharengus*) and Rainbow Smelt (*Osmerus mordax*) [27, 36, 37]. Cisco and these two non-indigenous species could have competed for food, given that they all are pelagic and rely heavily on zooplankton prey [27, 38]. Additionally, predation by adult Rainbow Smelt

on Cisco larvae has been proposed to have strong negative effects on Cisco populations in Lake Superior [39–43].

The dispersal history of Rainbow Smelt and Alewife indicates that they were likely expanding across the upper Great Lakes in the mid-twentieth century, although fishery independent accounts are limited. Between these two non-indigenous species, Rainbow Smelt arrived first, in the early twentieth century [44]. At that time, fishery managers intended to introduce Atlantic Salmon (*Salmo salar*) with the primary objective of establishing self-sustaining populations, so Rainbow Smelt was introduced as their food. Rainbow Smelt was stocked into numerous waterbodies in the region and the first report for an established population was in Crystal Lake in Michigan, USA in the 1910s. Later, Rainbow Smelt dispersed through connecting waterbodies and was first reported in Lake Michigan in 1923, in Lake Huron in 1925, and in Lake Superior in 1930 [45]. Rainbow Smelt became widespread across the upper Great Lakes in the 1930s [45], although the populations in Lakes Michigan and Huron were nearly exterminated in a huge winter kill event in 1942–1943 [46]. Alewife ultimately reached the upper Great Lakes after migrating from the Atlantic Ocean through the Great Lakes-St. Lawrence Seaway [47]. The species was first reported in Lake Huron in 1933, in Lake Michigan in 1949, and in Lake Superior in 1954 [47]. In the 1950s, both non-indigenous species were already present and potentially expanding [29, 30, 36], but spatiotemporal distributions based on fishery independent data have not been described.

In this study, our primary goal was to describe the distributions of Cisco in the upper Great Lakes in the mid-twentieth century, just prior to collapses of the Lakes Michigan and Huron fisheries and a severe decline in the Lake Superior fishery. Our secondary goal was to explore whether Alewife or Rainbow Smelt played a role in the decline of Cisco populations across the upper Great Lakes. We analyzed data from gill-net and bottom-trawl surveys collected by the U.S. Fish and Wildlife Service's research vessel R/V *Cisco* [20]. Although the R/V *Cisco* facilitated some of the earliest comprehensive studies on limnology [32, 48], benthic and pelagic invertebrate communities [49–53], and fish communities [54, 55], no one has previously analyzed these fish survey data quantitatively to describe or compare fish distributions across the Great Lakes for any species. Our results can provide new insights to fishery managers as to what distributions could be expected from recovered Cisco populations in Lakes Michigan and Huron and provide insights as to whether Alewife and Rainbow Smelt could have contributed to the historical decline of Cisco populations across the Great Lakes.

## Materials and methods

### Study area

The upper Great Lakes are bordered by the U.S. states of Minnesota, Wisconsin, Michigan, Illinois, and Indiana and the Canadian province of Ontario (Fig 1). Lake Superior has an area of 83,300 km$^2$ and a mean depth of 151 m. The R/V *Cisco* surveys covered eastern and western portions of Lake Superior in 1952–1953 and the eastern portion in 1959. Regions of interest surveyed in Lake Superior included the Apostle Islands, Keweenaw Bay, and Whitefish Bay. Lake Michigan includes three relatively distinct sub-basins: the main basin (area 52,720 km$^2$; mean depth 91 m), Green Bay (area 4,450 km$^2$; mean depth 17 m), and Grand Traverse Bay (area 720 km$^2$; mean depth 55 m). Lake Huron includes four relatively distinct sub-basins: the main basin (area 38,450 km$^2$; mean depth 70 m), Georgian Bay (area 13,990 km$^2$; mean depth 51 m), North Channel (area 4,010 km$^2$; mean depth 25 m), and Saginaw Bay (area 2,800 km$^2$; mean depth 8 m). Spring total phosphorus (TP) data indicate that the upper Great Lakes were more eutrophic historically (1950s–1970s) than today (2014–2019; Table 1).

**Table 1. Historical and contemporary levels of total phosphorus (TP) in the upper Great Lakes.**

| Lake | Sub-basin | Historical | | Contemporary | |
|------|-----------|------------|---|--------------|---|
| | | Period | TP (μg/L) | Period | TP (μg/L) |
| Superior | | 1953 | 11.2 (6.6) [33] | 2015–2019 | 2.6 (0.5)[a] |
| Michigan | Main basin | 1960–1961 | 15.1 (10.3) [34] | 2015–2019 | 2.7 (0.6)[a] |
| | Green Bay | 1963 | 75.3 (53.2) [56] | 2014 | 55.0 (30.9) [57] |
| | Grand Traverse Bay | 1973–1975 | 10.7 (6.7) [58] | 2015–2018 | 2.3 (1.9)[b] |
| Huron | Main basin | 1974 | 9.0 (5.2) [59] | 2015–2019 | 2.5 (0.5)[a] |
| | Georgian Bay | 1974 | 6.5 (4.9) [59] | 2017 | 2.5 (1.3)[c] |
| | North Channel | 1974 | 10.7 (7.3) [59] | 2017 | 3.5 (1.7)[c] |
| | Saginaw Bay | 1956 | 26.1 (15.9) [32] | 2017 | 17.5 (25.7) [60] |

Values are the mean and standard deviation (parentheses) of TP concentration in surface water in spring. Source data were from the earliest and latest comprehensive surveys.

[a]U.S. Environmental Protection Agency's Great Lakes Environmental Database, available at cdx.epa.gov

[b]Michigan's Water Chemistry Monitoring Program [61], data available at www.waterqualitydata.us

[c]Environment and Climate Change Canada's Great Lakes Water Quality Monitoring and Surveillance Data, available at open.canada.ca

## Data

Our source data were from gill-net and bottom-trawl surveys conducted by the R/V *Cisco* in the upper Great Lakes during 1952–1962 [62]. This dataset includes number of fish caught ("catch" in the source data), total weight of fish caught ("weight" in the source data), and information for fish surveys, such as fishing date, location, depth, gear, and effort [62]. In Lake Superior, the surveys were conducted in August 1952 and between May and October in 1953 and 1959. In Lake Michigan, the surveys were conducted from May to October 1952, in October 1953, from May to December 1954, from January to November 1955, and from April to November in 1960, 1961, and 1962. In Lake Huron, most of the surveys were conducted in the Saginaw Bay area from June to November 1956, but a small number of surveys were conducted in the North Channel in November 1952. No surveys were conducted in any of these lakes in 1957 and 1958.

These surveys were assigned to season based on relative differences in water temperature data collected by the R/V *Cisco* in the 1950s and 1960s [32–34]. In Lake Superior, surveys were assigned to spring (May through July 15), summer (July 16 through September 15), and fall (September 16 through October). In Lake Michigan, surveys were assigned to spring (April through June), summer (July through September), fall (October through December), and winter (January through March). In Lake Huron, surveys were assigned to spring (June only), summer (July through September), and fall (October and November).

We restricted our analyses to data for number of fish caught (hereafter, catch) of Ciscoes from standard gill-net surveys and catch of Ciscoes, Alewives, and Rainbow Smelts from standard bottom-trawl surveys. Gill-net survey data for Alewife and Rainbow Smelt were not analyzed because the two species were rarely caught in the surveys, likely owing to mesh sizes too large to entangle them. We note that the reported catch of these three species usually included only yearling and older (YAO) fish in the source data. For Cisco, the reported catch included Ciscoes with a total length of >152 mm, which should be mostly, if not entirely, YAO Ciscoes based on the historical length-at-age data [16, 63]. *Coregonus* fishes with a total length of <152 mm were not identified to species. Instead, they were enumerated and reported as one group of "Unidentified *Coregonus*," which we did not include in our analyses. For each of Alewife and Rainbow Smelt, the catch of fry was usually estimated and reported separately (e.g.,

"hundreds of fry"). However, in few cases, the reported catch of these two species was likely for the total of YAO and fry. In our analyses, we excluded catch data with a low mean weight of Alewife (<8 g [64]) or Rainbow Smelt (<3 g [40]), which indicated that catch was mostly consisted of fry.

The standard gill-net surveys conducted by the R/V *Cisco* employed gill nets made of multi-filament nylon and had a height of 1.8 m and used relatively consistent fishing methods across the upper Great Lakes in 1952–1962. In a standard survey, the gill nets could be set on the bottom or obliquely from the surface to the bottom and then lifted after one night in Lake Superior and Lake Huron or after up to 13 nights in Lake Michigan. A standard bottom-set gill-net lift included 1–12 gangs of gill nets. Each gang comprised a different number of nets, with a total length per gang ranging from 15 to 914 m and a mesh size (stretch) ranging from 25 to 152 mm. A standard oblique-set gill-net lift included one gang of gill nets, with a total length ranging from 56 to 366 m and a mesh size ranging from 25 to 89 mm.

The standard bottom-trawl surveys conducted by the R/V *Cisco* employed a range of different trawls but used very consistent fishing methods during daylight across the upper Great Lakes in 1952–1962. In total, 21 types of trawls made of tar-coated cotton were employed in these surveys. The headropes of these trawls ranged from 6.7 to 16.8 m in length. Most of these trawls had mesh sizes (stretch) of 38–64 mm at the mouth and 13 mm at the cod end. The other specifications of the trawls, such as design (e.g., semi-balloon or full balloon), mesh size of different part of the trawl, and net size, are given in the U.S. Geological Survey (USGS) data release [62]. In a standard survey, the trawl was lowered to the bottom, towed at a target speed of 3.4 km per hour for 2 to 66 minutes along a specified depth contour, and then lifted.

## Modeling gill-net Cisco catch

We used generalized additive models (GAMs) [65] to generate maps of predicted Cisco catch per unit effort (CPUE) by lake based on our gill-net data. This method has proven capable of generating predictive maps as accurate as traditional geostatistical (kriging) methods [66, 67]. We have used this method, too, in a previous study [68] to describe historical distribution of Cisco in Lake Michigan based on similar gill-net data collected in the 1930s [69].

Our GAM fitting was restricted to data collected from gill nets with mesh sizes of 38–76 mm from regions that were well-covered by the surveys. We excluded data collected from gill nets with mesh sizes <38 mm or >76 mm due to low catchability to these mesh sizes. This led to an exclusion of all data from five lifts across four stations in the North Channel, Lake Huron, made in November 1952, because only bottom-set gill nets with a mesh size of 25 mm were used. However, four of the five lifts caught Ciscoes (ranging 3–98 per lift), indicating that the species was abundant in this region. Across lakes, we excluded data from a total of eight spatially isolated stations (described below) because they offered limited value to generate predictive maps. In Lake Superior, we excluded data from three offshore stations. Only one lift of bottom-set gill nets was made at each of these three stations and no Ciscoes were caught. In Lake Michigan, we excluded data from the only station in Grand Traverse Bay, where only one lift of oblique-set gill nets was made, and it failed to catch Ciscoes. In Lake Huron, we excluded isolated data from four stations in the main basin that included 10 lifts of bottom-set gill nets, of which five caught 1–2 Ciscoes.

We further excluded some data in our GAM fitting due to small sample sizes. Only four lifts of oblique-set gill nets were made in Lake Superior in 1953, comprising 50 and 64 Ciscoes taken at two stations in spring and zero and three taken at two stations in summer. Winter surveys were only conducted at two stations in the southern main basin of Lake Michigan in 1955. These surveys included eight lifts of bottom-set gill nets, of which five caught 1–7

Ciscoes. In spring and summer surveys in Saginaw Bay, catch rates were too low to allow us to describe Cisco distribution, despite relatively large sample sizes. There were 13 lifts of oblique-set gill nets and four lifts of bottom-set gill nets across six stations in these Saginaw Bay surveys, but only three Ciscoes were caught in three different stations. The final gill-net dataset used in our GAM fitting comprised 128 lifts (515 gangs), 272 lifts (1,065 gangs), and 14 lifts (40 gangs) in Lake Superior, Lake Michigan, and Saginaw Bay, respectively.

Our GAMs were developed to predict CPUE for each lake because of the differences in gill-net fishing methods, lake morphology, and seasonal catch rates. We denoted our GAMs as Lake Superior, Lake Michigan, and Saginaw Bay as they were developed to generate maps of predicted Cisco CPUE in these waterbodies. We hypothesized that predicted Cisco CPUE and resultant distributions in the upper Great Lakes in the study period 1952–1962 were affected by fishing effort, season, bottom depth, and location, as they were shown to be important in our previous study on Cisco distribution in Lake Michigan in the 1930s [68]. Therefore, we used a backward-selection approach to determine the final predictors in each GAM, by starting with developing full models to include all potential predictors that could be derived from our data. Then, if not all predictors in the full model were significant at $\alpha = 0.05$, we eliminated the predictor with the highest p value. With the reduced model, we repeated the process until all predictors were significant.

These GAMs were built to model count data and the number of Cisco caught was used as the response variable. Accordingly, the predicted gill-net Cisco CPUE in this study was the GAM-predicted value of Cisco catch based on one unit of gill-net fishing effort, which we defined in the following subsection. For each GAM, we used a Tweedie error distribution with a natural logarithmic link function as the response variable has a count nature and could be over-dispersed [65]. The full model of each lake included two groups of predictors. The first group of predictors were for fishing effort adjustment specific to the gill-net fishing method used in the lake. The second group of predictors were for Cisco spatiotemporal distribution used in the selected GAM from the previous study [68]. For example, our Lake Superior full model was expressed as

$$\ln(Catch) \sim o(LNA) + s_m(Mesh) + Season + s_{sd}(Season \times Depth) + s_g(Lat, Lon) \quad (1)$$

where $\ln(\cdot)$ is the natural logarithmic link function of this GAM, *Catch* is the number of Cisco caught in a gang of gill nets, $o(LNA)$ means that *LNA*, the natural logarithm of the total mesh area of a gang of gill nets, is included as an offset variable which has a model coefficient constrained to one so that *Catch* can be modeled by using a Tweedie error distribution for count data [65], $s_m(Mesh)$ is a smoothing function of mesh size, *Season* is a categorical variable for the three seasons (spring, summer, and fall), *Depth* is a continuous variable of bottom-depth (the mean depth of the two ends of a gang of bottom-set gill nets or the station depth of oblique-set gill nets, $s_{sd}(Season \times Depth)$ means that a smoothing function of bottom depth is fitted for each season, and $s_g(Lat, Lon)$ is a Gaussian-process smoothing function [65] of latitude *Lat* and longitude *Lon*. In this full model, $o(LNA)$ and $s_m(Mesh)$ are predictors for fishing effort adjustment while *Season*, $s_{sd}(Season \times Depth)$, and $s_g(Lat, Lon)$ are predictors for spatiotemporal distribution.

In a similar manner, our Lake Michigan and Saginaw Bay full models were expressed as Eqs (2) and (3), respectively

$$\ln(Catch) \sim o(LNA) + Set + s_m(Mesh) + s_n(Night) + Basin + Season + s_{sd}(Season \times Depth) \\ + s_g(Lat, Lon) \quad (2)$$

$$\ln(Catch) \sim o(LNA) + Set + s_m(Mesh) + s_d(Depth) + s_g(Lat, Lon) \tag{3}$$

where *Set* is a categorical variable representing bottom-set or oblique-set gill nets, $s_n(Night)$ is a smoothing function of fishing duration in a unit of nights, *Basin* is a categorical variable used to represent the two sub-basins in Lake Michigan (Green Bay and the main basin), and $s_d(Depth)$ is a smoothing function of bottom depth. In the Lake Michigan full model, $o(LNA)$, *Set*, $s_m(Mesh)$, and $s_n(Night)$ are predictors for fishing effort adjustment while *Basin*, *Season*, $s_{sd}(Season \times Depth)$, and $s_g(Lat, Lon)$ are predictors for spatiotemporal distribution. In the Saginaw Bay full model, $o(LNA)$, *Set*, and $s_m(Mesh)$ are predictors for fishing effort adjustment while $s_d(Depth) + s_g(Lat, Lon)$ are predictors for spatiotemporal distribution. We did not include *Set* in Lake Superior full model because data from oblique-set gill nets were excluded in model fitting due to a small sample size of four lifts. We also did not include *Season* in Saginaw Bay full model as we only used fall survey data in model fitting. We only included $s_n(Night)$ in Lake Michigan full model because a fishing duration of one night was the standard in Lake Superior and Saginaw Bay surveys. We also only included *Basin* in Lake Michigan because there were no distinct sub-basins in Lake Superior and Saginaw Bay is already a sub-basin of Lake Huron.

Before starting our backward-selection process, we evaluated whether data from 1952–1956 and 1959–1962 could be combined and used in the fitting Lake Superior and Lake Michigan models. This step was necessitated because the spatial coverage of the gill net surveys was different (Fig 1), and there may have been shifts in Cisco distributions between the two periods in each of these lakes. To address this possibility, we modified the Lake Superior and Lake Michigan full models and fitted each of them to a subset of our gill-net data from stations that spatially overlapped in the two periods. For the Lake Superior full model, we added a categorical variable *Period*, which represents the periods 1952–1956 and 1959–1962, and fitted the modified model to data from stations east of Keweenaw Bay. For the Lake Michigan full model, we first removed *Basin* and *Set*, as the surveys were conducted only in the main basin with bottom-set gill nets in 1960–1962, and then added *Period* and fitted the modified model to data from stations in the main basin. We tested whether *Period* in each of the two models was significant at $\alpha = 0.05$. If *Period* was not significant, we built one model to generate a map for both periods. If *Period* was significant, we built one model to generate a map for each period.

We used the package "mgcv" version 1.8–36 [70] in R version 4.1.0 [71] for our GAM fitting and model selection. Following our previous study [68], we used the thin-plate regression spline as the smoothing basis, set the Gaussian-process smoothing function $s_g(Lat, Lon)$ to have a spherical correlation structure in which spatial auto-correlation was precluded if two points were separated by >90 km, and estimated parameters and smoothing functions by using the restricted-maximum-likelihood method for all GAMs. Refer to Wood [65] for the details about GAM fitting and tests of significance.

## Maps of predicted gill-net Cisco CPUE

We used the selected GAMs to estimate gill-net Cisco CPUE as the predicted Cisco catch based on predictors that represent one unit of gill-net fishing effort, which we defined as one lift of a gang of bottom-set gill nets, with a total mesh area of 100 m$^2$, a mesh size of 51 mm, and a fishing duration of one night. The bottom-set gillnetting with a fishing duration of one night was selected because it was the most used fishing method in our dataset. Most gill-net gangs had total mesh area of 100–200 m$^2$ in our dataset so the total mesh area of 100 m$^2$ was

selected for convenience. Although gill nets with mesh sizes of 51 mm and 64 mm were used most often in our dataset, the 51-mm mesh was selected because our preliminary analysis showed higher Cisco catchability to gill nets with this mesh size.

Because each of the Lake Superior, Lake Michigan, and Saginaw Bay full models included different predictors for fishing effort adjustment, different inputs associated with fishing effort could be required for each selected model to estimate gill-net Cisco CPUE. The Lake Superior full model only included $o(LNA)$ and $s_m(Mesh)$ as predictors for fishing effort adjustment because only data collected from gill nets set on the bottom and then lifted after one night were included in the model fitting. Thus, *Catch* in the Lake Superior model already represents number of Cisco caught from one lift of a gang of bottom-set gill nets, with a fishing duration of one night. Then, if the full Lake Superior model was selected, only a total gill-net mesh area of 100 m$^2$ and a mesh size of 51 mm were required as inputs for $o(LNA)$ and $s_m(Mesh)$, respectively, to estimate gill-net Cisco CPUE. In addition to $o(LNA)$ and $s_m(Mesh)$, the Lake Michigan full model also included *Set* and $s_n(Night)$ as predictors for fishing effort adjustment. Thus, if the full Lake Michigan model was selected, a fishing method of bottom-set gillnetting and a fishing duration of one night would be also required as inputs for *Set* and $s_n(Night)$, respectively, to estimate gill-net Cisco CPUE. Similarly, in addition to $o(LNA)$ and $s_m(Mesh)$, the Saginaw Bay full model also included *Set* as a predictor for fishing effort adjustment. This is because a fishing duration of one night was the standard in Saginaw Bay surveys. If the full Saginaw Bay model was selected, a fishing method of bottom-set gillnetting would be also required as the input for *Set* to estimate gill-net Cisco CPUE.

To generate maps of predicted gill-net Cisco CPUE, we used geographic information system (GIS) software ArcGIS version 10.8 (http://www.esri.com/software/arcgis/). The GIS layers of predicted gill-net Cisco CPUE were based on the predictions of selected GAMs. In addition to one unit effort defined above, we obtained input data for GAM predictions, including latitude, longitude, and bottom depth, from the Great Lakes Aquatic Habitat Framework's [17] public domain spatial database (https://glahf.org/data/). To avoid over-extrapolation, we restricted our predictive maps to the fishery management statistical districts [72] that encompass the survey stations for our gill-net data. In Lake Superior, we restricted our predictions in 1952–1956 to all USA statistical districts and Ontario statistical district OS-7 and in 1959–1962 to statistical districts MS-3 to 6 and OS-7. In Lake Michigan, we restricted our predictions in 1952–1956 to all statistical districts but MM-4 (Grand Traverse Bay), and in 1959–1962 to the statistical districts in main basin. As Saginaw Bay is part of the statistical district MH-4, we further restricted our predictions to fishery management statistical grids [17] that encompass the bay, including grids 1308–1309, 1408–1411, 1506–1509, 1606–1609, and 1707–1708. Refer to the Great Lakes Aquatic Habitat Framework's [17] public domain spatial database for the maps of fishery management statistical districts and grids in the Great Lakes.

## Maps of observed bottom-trawl CPUE

We also used ArcGIS version 10.8 to generate maps of observed bottom-trawl CPUE for Cisco, Alewife, and Rainbow Smelt. Effort for CPUE was approximated by the width of the headrope multiplied by the distance towed. Thus, CPUE is assumed to be proportional to actual catch per area swept by the trawl. To maximize comparability of CPUE, at least within each lake, we further restricted our data to the most-used trawls. In the period 1952–1956, most of the surveys across all three lakes employed four-seam trawls (i.e., the trawl consists of four nets joined laterally like a box) with headrope lengths of 9.4–11.9 m. In the period 1959–1962, all Lake Superior surveys still employed four-seam trawls with headrope lengths of 9.4–11.9 m, but most of the Lake Michigan surveys employed two-seam trawls (i.e., the trawl

consists of an upper net and a lower net joined laterally like a bag) with a headrope length of 15.2 m. In total, our maps of CPUE for 1952–1956 were based on 259, 73, and 42 tows for Lake Superior, Lake Michigan, and Saginaw Bay, respectively, and for 1959–1962 on 90 and 216 tows for Lake Superior and Lake Michigan, respectively. No bottom-trawl surveys were conducted outside of Saginaw Bay, Lake Huron, in the period 1952–1956.

## Results

### Modeling gill-net Cisco catch

Our evaluations showed that data from the periods 1952–1956 and 1959–1962 could not be combined for the Lake Superior model (*Period*: p < 0.001) but could be combined for the Lake Michigan model (*Period*: p = 0.204). The full Lake Superior model was selected for the first period (included data from 1952 and 1953) and explained 60% of the deviance (Table 2; Fig 2A). The model showed that Cisco had the highest catchability to gill nets with a mesh size of around 51 mm (Fig 3A). The model also showed that Cisco catch was higher in spring than in summer and fall (Fig 3B) and decreased with increasing bottom depth in all three seasons, although the decreasing trend was less significant in spring (Fig 3C–3E). For the second period (based on 1959 data), the full Lake Superior model was again selected, and it explained 59% of the deviance (Table 2; Fig 2B). The model showed that Cisco had the highest catchability to gill nets with a mesh size of between 51 and 64 mm (Fig 3F). Similar to the 1952–1953 period, the 1959 model showed that Cisco catch was higher in spring than in summer and fall (Fig 3G), and decreased with increasing bottom depth in spring and fall (Fig 3H and 3J). In summer, however, the 1959 Cisco catch was more non-linear, peaking at a bottom depth of around 30 m (Fig 3I).

Based on the fit to data from 1952–1962, the selected Lake Michigan model included every predictor in the full model but *Set* (p = 0.097) and explained 71% of the deviance (Table 2; Fig 2C). The model showed that Cisco had the highest catchability to gill nets with a mesh size of around 51 mm (Fig 4A). The model also showed that Cisco catch increased log-linearly with fishing duration, corresponding to about a 15% increase in Cisco catch per additional night of fishing (Fig 4B). Between sub-basins, that model showed that Cisco catch was significantly higher in Green Bay than in the main basin (Fig 4C). Across seasons, the model showed that Cisco catch increased as the season progressed from spring to fall (Fig 4D) and had variable bottom-depth distributions: in spring, Cisco catch generally decreased with increasing bottom

**Table 2. The p values of intercepts and selected predictors in the generalized additive models fitted to gill-net data.**

|  | Model (data period) | | | |
|---|---|---|---|---|
|  | **Lake Superior (1952–1953)** | **Lake Superior (1959)** | **Lake Michigan (1952–1962)** | **Saginaw Bay (1956)** |
| Intercept | <0.001 | <0.001 | <0.001 | <0.001 |
| $s_m(Mesh)$ | <0.001 | <0.001 | <0.001 | 0.005 |
| $s_n(Night)$ |  |  | <0.001 |  |
| Basin |  |  | <0.001 |  |
| Season | 0.026 | 0.016 | 0.002 |  |
| $s_d(Depth)$ |  |  |  | 0.001 |
| $s_{sd}(Spring \times Depth)$ | 0.031 | 0.003 | <0.001 |  |
| $s_{sd}(Summer \times Depth)$ | <0.001 | <0.001 | 0.036 |  |
| $s_{sd}(Fall \times Depth)$ | 0.047 | 0.041 | 0.022 |  |
| $s_g(Lat, Lon)$ | <0.001 | <0.001 | <0.001 | <0.001 |

Refer to Methods for predictor definitions.

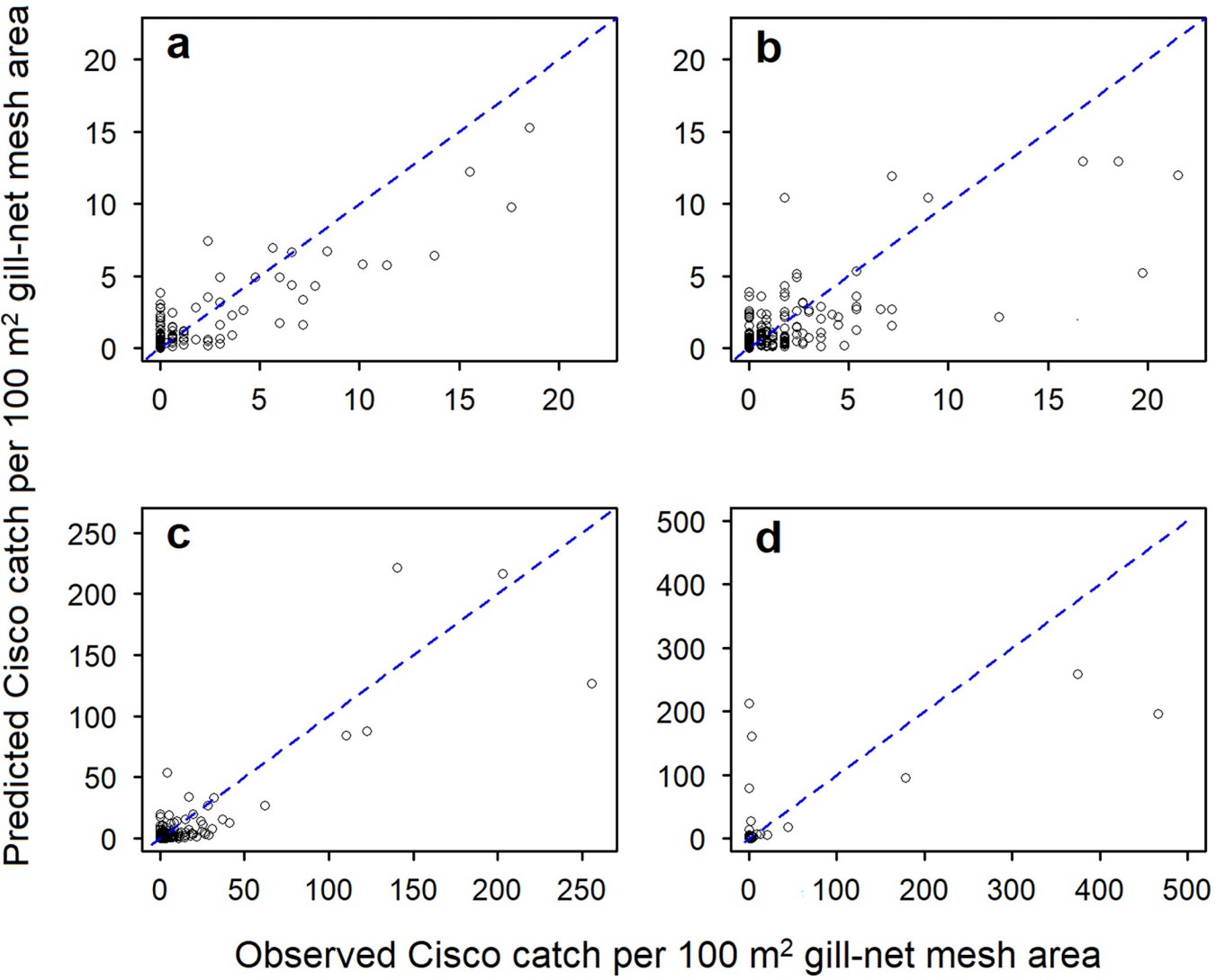

**Fig 2.** Observed and predicted Cisco catch per 100 m$^2$ gill-net mesh area for (a) Lake Superior in 1952–1953, (b) Lake Superior in 1959, (c) Lake Michigan in 1952–1962, and (d) Saginaw Bay in fall, 1956. Circles represent data points and dashed line represents a 1:1 relationship.

depth (Fig 4E); in summer, catch peaked at a bottom depth of around 50 m but was also relatively high out to bottom depths of 160 m (Fig 4F); in fall, catch was highest in the shallowest waters and then exhibited a second peak at a bottom depth of about 160 m (Fig 4G).

The selected Saginaw Bay model, based on the fit to data from fall 1956, included every predictor in the full model but *Set* (p = 0.071) and explained 67% of the deviance (Table 2; Fig 2D). The model showed that Cisco had the highest catchability to gill nets with a mesh size of around 64 mm (Fig 5A) and Cisco catch decreased log-linearly with bottom depth (Fig 5B).

## Maps of predicted gill-net Cisco CPUE

Our maps (Fig 6) of predicted gill-net Cisco CPUE showed the importance of embayment and shallow-water habitats to Cisco. Across all seasons, the model consistently predicted high CPUE in Keweenaw Bay and Whitefish Bay, Lake Superior, and in Green Bay, Lake Michigan, with the highest predicted CPUE occurring in northern Green Bay. The predicted CPUE was

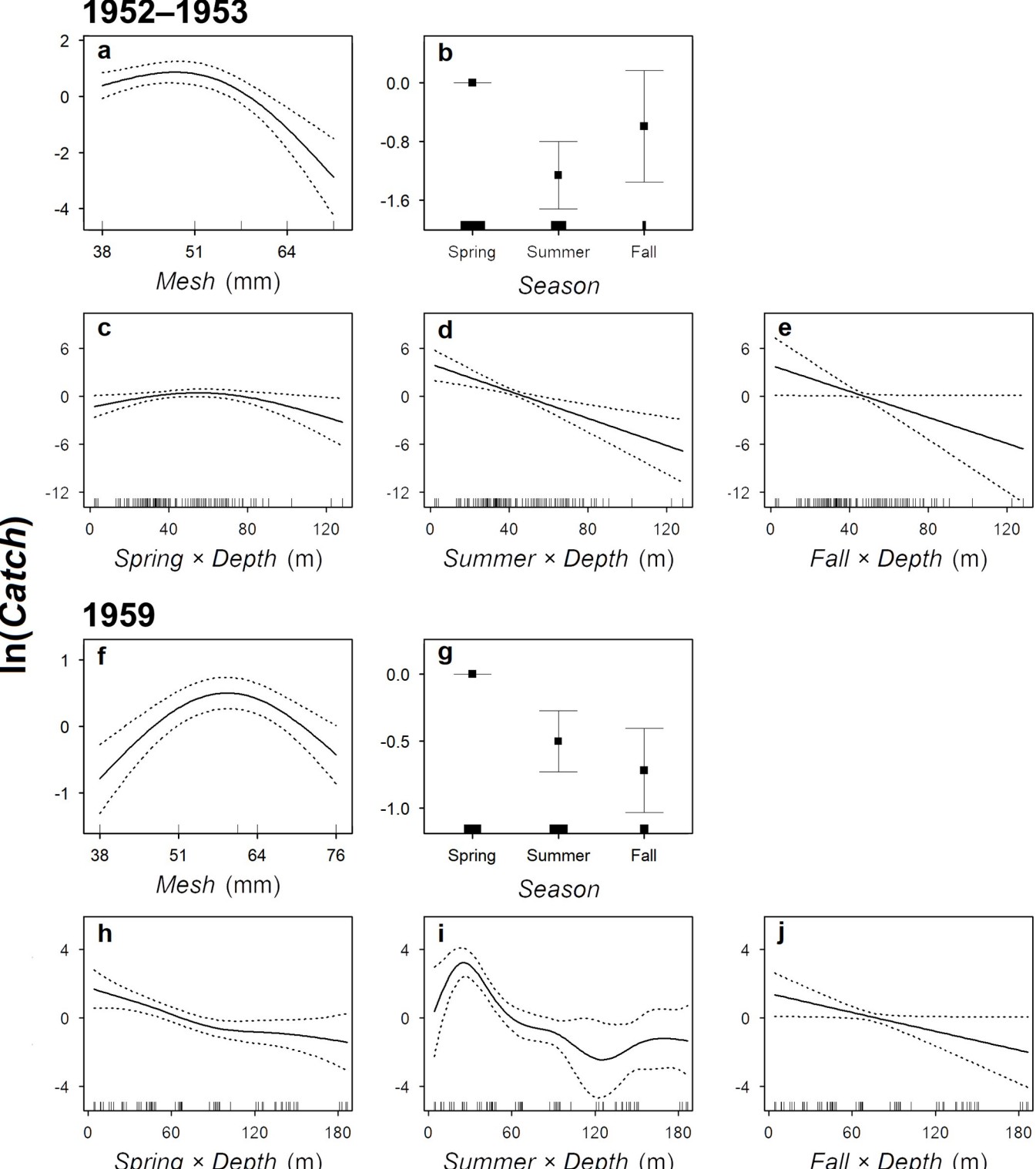

**Fig 3. The relationships between predicted number of Cisco caught (*Catch*) and predictors in the selected generalized additive models fitted to gill-net data from Lake Superior.** *Mesh* is a continuous variable representing gill-net mesh size (stretch). *Season* is a categorical variable representing spring, summer, and fall. *Depth* is a continuous variable representing bottom depth of which a smoothing function was fitted for each season. In panels for the smoothing functions (a, c–e, f, and h–j), solid lines represent the predicted values, dotted lines represent +/− one standard error, and ticks above the horizontal axis represent the distribution of data points. In the panels for categorical variables (b and g), points represent the predicted values, error bars represent +/− one

standard error, and ticks above the horizontal axis represent relative sample size. The final predicted values of *Catch* still need to be adjusted for spatial variability (i.e., latitude and longitude), offset values (i.e., gill-net mesh area), and intercepts of the models. At the same scale of ln(*Catch*), the model fitted to 1952–1953 data has an intercept of −4.855 (standard error of 0.217) and the model fitted to 1959 data has an intercept of −5.392 (standard error of 0.143).

also high in Saginaw Bay, Lake Huron, in fall, particularly along the southern shore in the outer bay region. Note that we could not generate predictive maps for Saginaw Bay in spring and summer due to low Cisco catch rates. Outside of these major embayments, the predicted CPUE was generally higher in shallower waters close to the shore. However, the difference

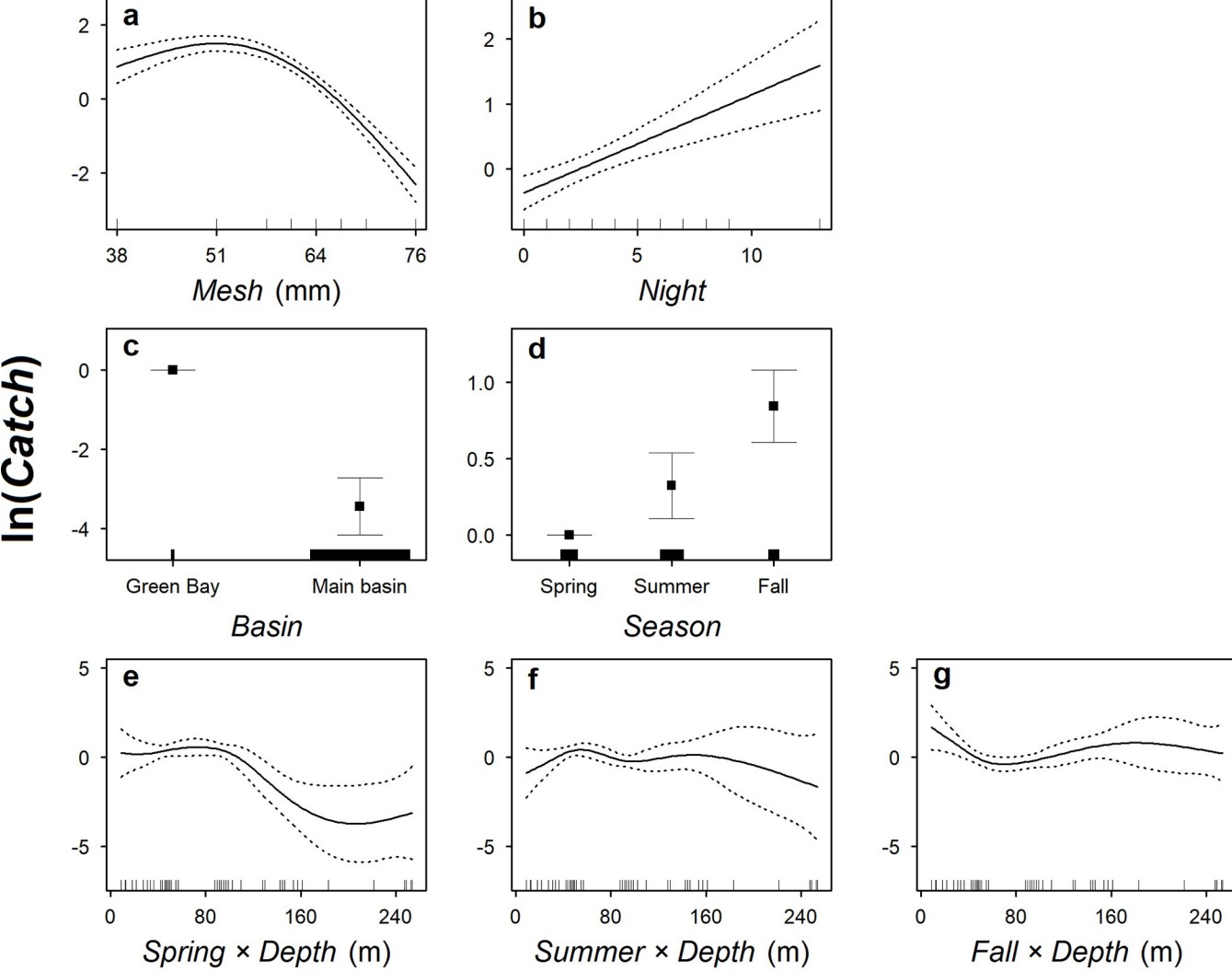

**Fig 4. The relationships between predicted number of Cisco caught (*Catch*) and predictors in the selected generalized additive model fitted to gill-net data collected from Lake Michigan in 1952–1962.** *Mesh* is a continuous variable representing gill-net mesh size (stretch). *Night* is a continuous variable number of nights fishing (*Night* = 0 means that gill nets were set in the morning but lifted before midnight). *Basin* is a categorical variable representing the two sub-basins, Green Bay and the main basin, in Lake Michigan. *Season* is a categorical variable representing spring, summer, and fall. *Depth* is a continuous variable representing bottom depth of which a smoothing function was fitted for each season. In panels for the smoothing functions (a–b and e–g), solid lines represent the predicted values, dotted lines represent +/− one standard error, and ticks above the horizontal axis represent the distribution of data points. In the panels for categorical variables (c and d), points represent the predicted values, error bars represent +/− one standard error, and ticks above the horizontal axis represent relative sample size. The final predicted values of *Catch* still need to be adjusted for spatial variability (i.e., latitude and longitude), offset values (i.e., gill-net mesh area), and the intercept in the model. At the same scale of ln(*Catch*), the model has an intercept of −2.435, with a standard error of 0.703.

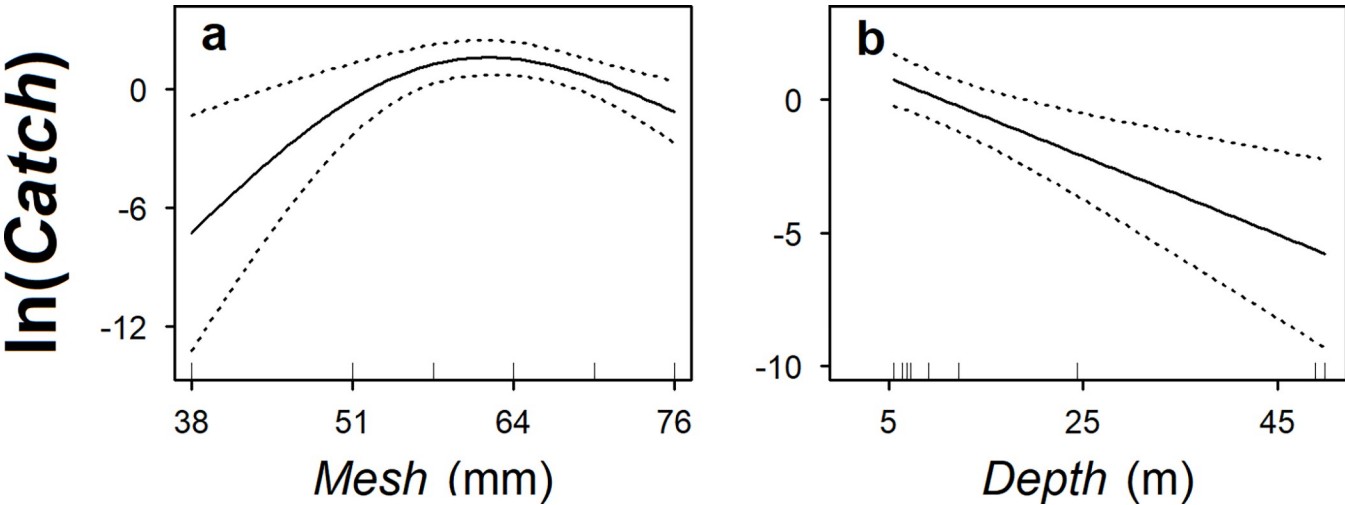

**Fig 5. The relationships between predicted number of Cisco caught (*Catch*) and predictors in the selected generalized additive model fitted to gill-net data collected from Saginaw Bay in fall, 1956.** *Mesh* is a continuous variable representing gill-net mesh size (stretch) and *Depth* is a continuous variable representing bottom depth. Solid lines represent the predicted values, dotted lines represent +/− one standard error, and ticks above the horizontal axis represent the distribution of data points. The final predicted values of *Catch* still need to be adjusted for spatial variability (i.e., latitude and longitude), offset values (i.e., gill-net mesh area), and the intercept in the model. At the same scale of ln(*Catch*), the model has an intercept of −4.219, with a standard error of 0.464.

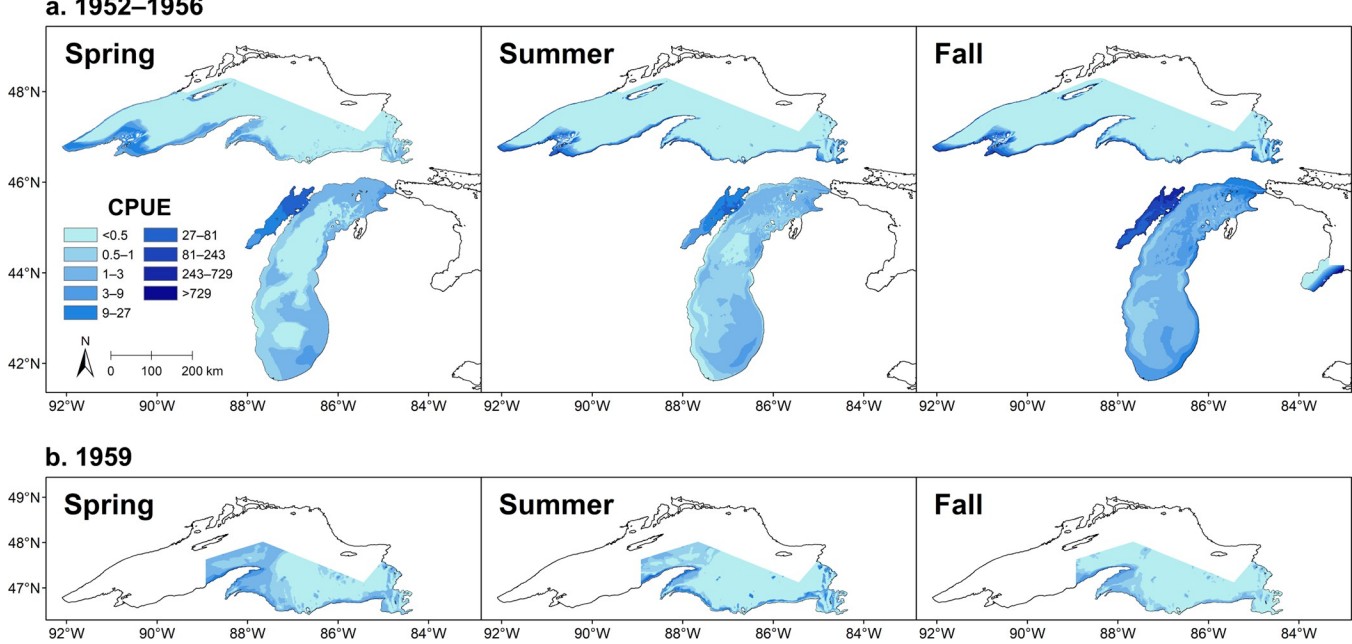

**Fig 6.** Maps of predicted gill-net Cisco catch per unit effort (CPUE) in (a) the upper Great Lakes in 1952–1956 and (b) Lake Superior in 1959. The CPUE was predicted from selected generalized additive models fitted to gill-net data collected by the R/V *Cisco*. A unit of effort was defined as one lift of a gang of bottom-set gill nets, with a total mesh area of 100 m², a mesh size of 51 mm, and a fishing duration of one night. The predictions were restricted to areas covered by R/V *Cisco* surveys. Note that our model selection results showed that the predicted Cisco CPUE in Lake Michigan would not be significantly different between the periods 1952–1956 and 1959–1962; accordingly, the Lake Michigan maps in Fig 6A represent the distributions of gill-net Cisco CPUE over the whole study period 1952–1962. Maps were generated with geographic information system (GIS) software ArcGIS version 10.8 (http://www.esri.com/software/arcgis/). The GIS layers of predicted Cisco gill-net CPUE were generated in this study. The GIS layer of the Great Lakes outline was obtained from the Great Lakes Aquatic Habitat Framework's [17] public domain spatial database (https://glahf.org/data/).

between predicted CPUE in shallow and deep waters was larger in Lake Michigan than in Lake Superior, especially in summer and fall.

Within Lake Superior, our maps revealed similar spatiotemporal distributions of the predicted CPUE in areas where the surveys overlapped during 1952–1953 and 1959, although predicted CPUE was generally higher in 1959. However, the predicted CPUE in 1952–1953 was higher in the western Lake Superior area that was not covered by 1959 surveys, particular in the Apostle Islands region.

## Maps of observed bottom-trawl CPUE

In 1952–1956, our maps of observed bottom-trawl Cisco CPUE showed that the CPUE was generally higher in Lake Superior than in either Lake Michigan, where the CPUE was frequently zero, or Saginaw Bay, where no Ciscoes were caught (Fig 7; Table 3). Within Lake

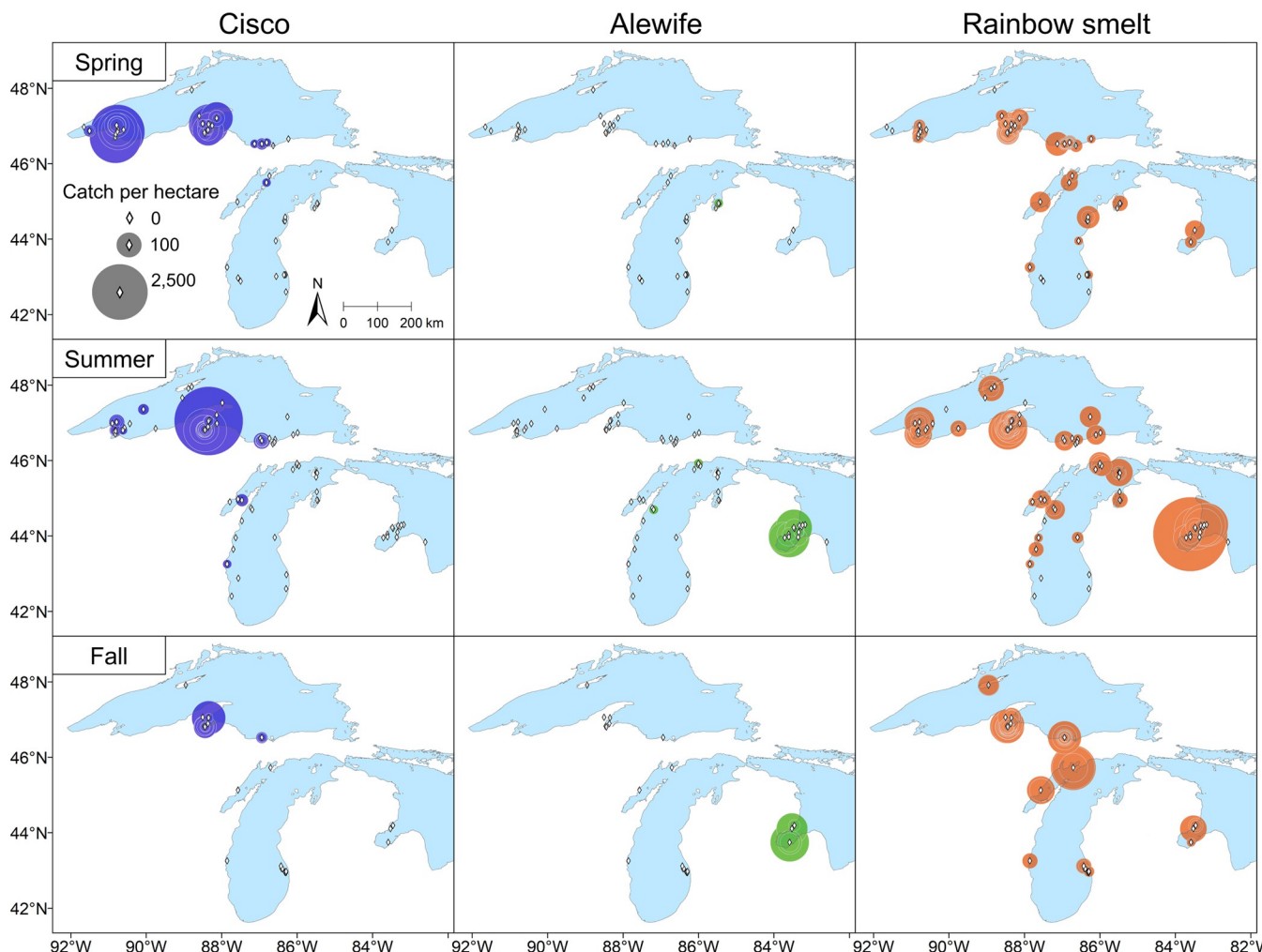

**Fig 7. Maps of observed bottom-trawl catch per unit effort (CPUE) for Cisco, Alewife, and Rainbow Smelt in the upper Great Lakes, 1952–1956.** The CPUE is expressed as catch per hectare of area swept by the trawl. Data were collected by four-seam trawls with headrope lengths of 9.4–11.9 m. Each diamond represents a survey station. Each circle, centered at a diamond, represents the CPUE at that station, and its area is proportional to the square root of CPUE. Each diamond without a circle around it represents a zero CPUE at that station. The maps were generated with geographic information system (GIS) software ArcGIS version 10.8 (http://www.esri.com/software/arcgis/). The GIS layers of observed bottom-trawl CPUE were generated in this study, while the GIS layer of the Great Lakes outline was obtained from the Great Lakes Aquatic Habitat Framework's [17] public domain spatial database (https://glahf.org/data/).

**Table 3. Percentages of non-zero values in observed bottom-trawl catch per unit effort (CPUE) for Cisco, Alewife, and Rainbow Smelt in the upper Great Lakes.**

| Lake | Season | N | Non-zero CPUE (%) | | |
|------|--------|---|-------|---------|--------------|
| | | | Cisco | Alewife | Rainbow Smelt |
| **1952–1956** | | | | | |
| Superior | Spring | 70 | 24% | 0% | 61% |
| | Summer | 153 | 28% | 0% | 58% |
| | Fall | 36 | 28% | 0% | 86% |
| Michigan | Spring | 26 | 4% | 4% | 35% |
| | Summer | 28 | 7% | 7% | 61% |
| | Fall | 19 | 0% | 0% | 63% |
| Huron (Saginaw Bay) | Spring | 2 | 0% | 0% | 100% |
| | Summer | 26 | 0% | 38% | 58% |
| | Fall | 14 | 0% | 50% | 36% |
| **1959–1962** | | | | | |
| Superior | Spring | 10 | 0% | 0% | 70% |
| | Summer | 71 | 0% | 0% | 35% |
| | Fall | 9 | 0% | 0% | 22% |
| Michigan | Spring | 58 | 36% | 45% | 50% |
| | Summer | 124 | 29% | 31% | 56% |
| | Fall | 34 | 35% | 85% | 65% |

Superior, the maps showed relatively higher bottom-trawl Cisco CPUE in Keweenaw Bay and Apostle Islands region, which is consistent with the distributions of our predicted gill-net Cisco CPUE (Fig 6). The Alewife CPUE was relatively higher in Saginaw Bay than in either Lake Michigan, where the CPUE was frequently zero, or Lake Superior, where no Alewives were caught. In contrast, the Rainbow Smelt CPUE revealed a much broader distribution in the upper Great Lakes. Combining our results of predicted gill-net CPUE and observed bottom-trawl CPUE in this period (Figs 6 and 7), we found that the likelihood of spatiotemporal overlap between Rainbow Smelt and Cisco was relatively high across the upper Great Lakes. However, the spatiotemporal overlap between Alewife and Cisco was only likely in Saginaw Bay in fall.

In 1959–1962, our maps of observed bottom-trawl Cisco CPUE showed that no Ciscoes were caught in Lake Superior while Ciscoes were caught throughout the main basin of Lake Michigan (Fig 8; Table 3). By this period, however, Alewife were broadly distributed throughout Lake Michigan, as indicated by their relatively high CPUE compared to 1952–1956. Likewise, Rainbow Smelt CPUE revealed a broad distribution in 1959–1962 in both Lake Superior and Lake Michigan, similar to 1952–1956. Combining our results of predicted gill-net CPUE and observed bottom-trawl CPUE in this period (Figs 6 and 8), we found that the likelihood of spatiotemporal overlap between Alewife, Rainbow Smelt, and Cisco was relatively high in the main basin of Lake Michigan, and the likelihood of spatiotemporal overlap remained high between Rainbow Smelt and Cisco in Lake Superior.

## Discussion

### Cisco distributions in 1952–1962

By analyzing fishery-independent gill-net survey data for 1952–1962 from surveys conducted by the R/V *Cisco*, we found that Cisco was most abundant in embayments and shallow waters across the upper Great Lakes. According to our predictive maps, major embayments including

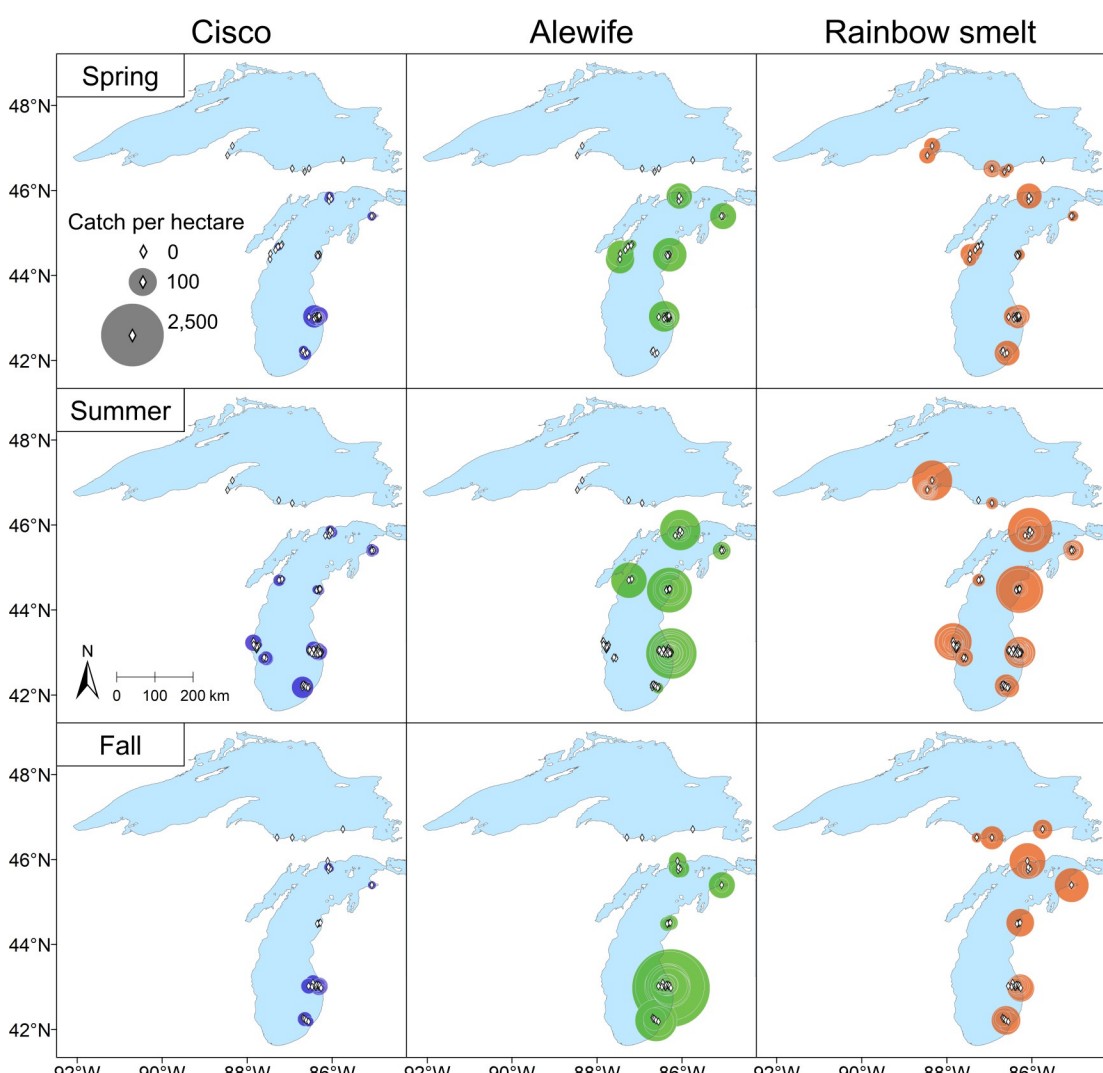

**Fig 8. Maps of observed bottom-trawl catch per unit effort (CPUE) for Cisco, Alewife, and Rainbow Smelt in Lake Superior and Lake Michigan, 1959–1962.** The CPUE is expressed as catch per hectare of area swept by the trawl. Lake Superior data were collected from four-seam trawls with headrope lengths of 9.4–11.9 m, while Lake Michigan data were collected from two-seam trawls with a headrope length of 15.2 m. Each diamond represents a survey station. Each circle, centered at a diamond, represents the CPUE at that station, and its area is proportional to the square root of CPUE. Each diamond without a circle around it represents a zero CPUE at that station. The maps were generated with geographic information system (GIS) software ArcGIS version 10.8 (http://www.esri.com/software/arcgis/). The GIS layers of observed bottom-trawl CPUE were generated in this study, while the GIS layer of the Great Lakes outline was obtained from the Great Lakes Aquatic Habitat Framework's [17] public domain spatial database (https://glahf.org/data/).

Keweenaw Bay and Whitefish Bay of Lake Superior, Green Bay of Lake Michigan, and Saginaw Bay of Lake Huron were productive for Cisco. Although the predicted gill-net Cisco CPUE in Saginaw Bay in fall was relatively high across the Great Lakes, these predictions were made based on data collected in 1956 surveys. However, nearly no Ciscoes were caught in spring and summer in the 1956 surveys, which is a sign of Cisco local extirpation. Saginaw Bay and Green Bay historically had been the most important fishing grounds in the upper Great Lakes [73, 74], but the year 1956 happened to be the first of a string of years when the Saginaw Bay fishery, in retrospect, was making its last gasp. Commercial catch in Saginaw Bay amounted to only 28 t in 1956, an amount that would have been in the rounding error of the peak catch of

2,400 t made in 1916; by 1963 the catch was nil [74]. Outside of these major embayments, the Apostle Islands region in Lake Superior was also productive of Cisco, where predicted gill-net CPUE was on par with those predicted for Keweenaw Bay and Whitefish Bay in 1952–1953. With only one sampling event in Grand Traverse Bay, Lake Michigan, we cannot infer anything about its Cisco abundance. The average commercial catch in Grand Traverse Bay during our study period was only about 4 t per year, which was 3% of the historical high of 1930 [75], indicating low Cisco abundance in the bay.

Our datasets for Lakes Superior and Michigan were extensive enough to reveal that Cisco seasonally inhabited different bottom depths in these two lakes. In Lake Superior, our analysis of gill-net survey data showed that Cisco stayed mostly in waters <80 m in bottom depth throughout the year. Our results for distributions in spring and summer are consistent with several early studies [76–78]. Selgeby and Hoff [78] reported that Cisco during 1958–1974 was abundant at bottom depths <70 m during spring and summer across Lake Superior. As Cisco will actively avoid water temperatures greater than 17˚C [79–81], their results indicate that these habitats in Lake Superior typically remained cool enough for Cisco during summer. Our results for bottom-depth distributions in fall are consistent with Dryer [77] for the Apostle Islands region of western Lake Superior during 1958–1963. However, based on data collected between November 16 and December 20 in western Lake Superior during 1958–1961, Dryer and Beil [76] reported that Cisco, when spawning, was abundant at all bottom depths out to 100 m, with the highest abundance occurring at around 100 m. Likewise, Selgeby and Hoff [78] reported Cisco aggregated at bottom depths of 90–120 m in late September through December across Lake Superior during 1958–1974, although the aggregations occurred late in fall towards the end of the spawning period, when Cisco were known to be in deeper water [76]. The fall R/V *Cisco* surveys in Lake Superior did not detect Cisco at bottom depths of 90–120 m. These surveys ended before late October, indicating they missed the late-fall spawning period and the associated aggregations of Cisco spawning in deep water.

Our fall comparisons of bottom-depth distributions between Lakes Superior and Michigan are somewhat confounded by the lack of sampling during November and December when Cisco was spawning in Lake Superior. In Lake Michigan, our data indicate a somewhat uniform distribution out to bottom depths of 250 m, which is inconsistent with other studies. For example, Smith [81] reported that Cisco in Green Bay, where most of the spawning in Lake Michigan took place and where our samples are most plentiful, spawned mostly at bottom depths <20 m, although spawning "takes place over practically all [bottom] depths and in all sections of the bay." Similarly, Koelz [15] reported that Cisco was rarely caught in waters with bottom depths >45 m in the 1920s. Likewise, our previous study [68] showed that Cisco abundance in Lake Michigan in the 1930s could be very low throughout the year in waters with bottom depths >70 m. In contrast to these historical data, the R/V *Cisco* surveys found that Cisco could be abundant in very-deep waters (>150 m bottom depths) in summer and fall. This conundrum may be explained by changes in the Cisco populations in the main basin of Lake Michigan from the early twentieth century [15, 68] to our study period, 1952–1962. Smith [55] inferred hybridization between the typical *artedi* form of Cisco and forms of deepwater ciscoes, based on morphological examination of fish collected in 1960–1961—hybridization among forms of Cisco is reasonably well documented [82]. Part of what underlies the broader bottom-depth distribution in fall in our study period was ripe female Cisco (based on limited biological data [68]) caught at several R/V *Cisco* survey stations at a bottom depth of 146 m in November 1961. Given the unexpected results of deeper fall distributions, future research involving morphological and genetic analyses are needed to test this hybridization hypothesis.

Although not an objective of this study, our predictive mapping indicated that inputs of TP from the dominant tributaries to Green Bay (Fox River) and Saginaw Bay (Saginaw River) had

only regional effects on Cisco habitat utilization within these embayments. While the upper Great Lakes were more eutrophic in the mid-twentieth century than today (Table 1), the stress of habitat degradation appeared to have been limited to regions adjacent to these major nutrient inputs, i.e., the lower bay region of Green Bay [56] and inner Saginaw Bay [32]. Our predictive maps showed that the highest Cisco abundances occurred in northern (upper) Green Bay and along the southern shoreline of outer Saginaw Bay, not in the shallower, more eutrophic regions near the major riverine sources of nutrients. Our predictive map for Green Bay was consistent with findings in our previous study on Cisco distribution in the 1930s [68], indicating that habitat degradation in lower Green Bay had long-standing effects on Cisco distributions. Given that fishery productivity remained relatively high for more than 10 years before the mid-1950s in both Green Bay and Saginaw Bay [20], our results indicate that high catches were not based solely on excessive nutrient input to these embayments.

### The role of non-indigenous species in the decline of Cisco populations

To what extent were Alewife and Rainbow Smelt abundant enough to be considered possible causes of the decline of Cisco populations in the upper Great Lakes during 1952–1962? Based on our analysis of gill-net and bottom-trawl survey data collected by the R/V *Cisco* during 1952–1956, we found that the spatiotemporal overlap between Rainbow Smelt and Cisco likely occurred across the upper Great Lakes but that overlap between Alewife and Cisco was likely limited to Saginaw Bay. By 1959–1962, however, Alewife were also widely distributed within Lake Michigan indicating likely spatiotemporal overlap with Cisco. Hence, to the extent that negative interactions with non-native planktivorous species influenced the decline of Cisco populations across the upper Great Lakes in the 1950s, the timing of overlap with Rainbow Smelt provides better evidence than the timing of overlap with Alewife. At the same time, any potential recovery of Cisco in the early 1960s could have been inhibited by Alewife in Lakes Michigan and Huron. The potential interaction between Rainbow Smelt and Cisco have been well-studied in Lake Superior [see review in 83], but not in Lakes Michigan and Huron as the Cisco population collapsed before studies could be launched. Thus, additional research on the potential interactions between Rainbow Smelt and Cisco would be helpful to inform ongoing Cisco restoration in these two lakes.

### Data biases

The majority of data we analyzed were collected by using bottom fishing gears (i.e., bottom-set gill net and bottom trawl), which likely under sample pelagic-oriented fishes like Cisco over deep waters [84–86]. For example, Dryer and Beil [76] reported that bottom-set gill nets could catch Cisco efficiently only when they are spawning. Therefore, using cisco catch rates of bottom-fishing gears as an index of cisco abundance indices are likely biased low. As our focus here was the distribution and relative abundance of Cisco, our findings would be biased if Cisco encountered the bottom fishing gears at different rates at stations with, for example, different bottom depths. However, our model selection results showed that bottom-set and oblique-set gill nets caught Cisco at comparable rates in Lake Michigan and Saginaw Bay, which indicates that abundance indices based on Cisco catch rates either at the bottom or in the water column would show the same patterns of Cisco distributions. If, however, cooler water temperatures near the surface of Lake Superior throughout the year allow Cisco to be more pelagic-oriented in Lake Superior than in Lakes Michigan and Huron [15, 76], our distributional findings in Lake Superior may not be as robust as in the other lakes where oblique gill net sets were also used.

While the visual identification of ciscoes is a well-known challenge, misidentification of Cisco as another species and vice versa should be less an issue in our data. Our confidence

arises from the assumption that researchers collecting these data were very experienced in identifying ciscoes, as this fish assemblage was a research priority at that time given their importance to fisheries [87]. Thus, although the misidentification of Cisco could have occurred in our source data, it unlikely occurred in a systematic manner to have had major effects on our results.

## Management implications

Across the upper Great Lakes, Cisco has received increased management attention. In Lake Superior, Cisco is widespread and fishery managers are concerned by its low and inconsistent recruitment [88]. In Lake Michigan, the abundance of Cisco has been increasing especially in northern waters [24] but more research is needed to understand its seasonal and spatial distribution. Interestingly, studies have also shown that contemporary Cisco and historical Cisco are different in morphology [16] and trophic position [89]. In Lake Huron, Cisco is most abundant in North Channel and Georgian Bay [16] but more research is needed to understand their seasonal distributions and trophic positions in different locations.

Restoring Cisco to a "significant level" remains an objective of fishery management in Lake Huron [90], and this priority was elevated following the collapse of the Alewife population around 2003 [91]. Since 2018, a reintroduction program for Cisco, focused on Saginaw Bay, has been underway and involves stocking one million fingerlings per year for 10 years. The source brood stock in northern Lake Huron, however, has different morphology than the historically dominant typical *artedi* form of Cisco in Saginaw Bay [15, 25, 26]. The current plan to evaluate recruitment of stocked Ciscoes includes bottom trawling [92]. However, data from the R/V *Cisco* showed that the catchability of Cisco to bottom trawling could be low, although trawls with long headropes may catch Cisco better. These results reinforce the current Saginaw Bay, multiple-gear (targeting multiple life history stages) evaluation approach [92].

Our findings can help fill two knowledge gaps confronting fishery management. First, for Lake Michigan, our results provide managers with benchmarks for the seasonal distribution of a recovered Cisco population. These results indicate that Cisco abundances will be highest in the embayments but also provides evidence that Cisco may occupy deep waters of the main basin in all seasons. However, the taxonomy of Ciscoes in our deep-water samples from summer and fall needs to be resolved to determine whether they were hybrids, as speculated by Smith [55], the lead scientist of R/V *Cisco* surveys. Similarly, if the recovering Cisco in Lake Michigan and the stocked Cisco in Saginaw Bay are subsequently determined to be taxonomically different from the historically dominant Cisco in these waterbodies, as recently proposed [16, 25, 26], our maps will provide a basis for assessing how much of the niche formerly occupied by historical Cisco is colonized by contemporary Cisco.

Second, our findings are consistent with research in Lake Superior, which has implicated Rainbow Smelt as a contributing factor in the decline of Cisco in this lake [39–43]. As restoration programs continue in Lake Huron or if conservation or restoration programs begin in Lake Michigan, a key step in the development of those programs will be a "threats assessment," which would identify the key stressors that led to the decline or local extirpation of Cisco. Our research does not rule out the importance of other possible stressors, such as overfishing, habitat degradation owing to eutrophication, and damming of rivers, but does highlight the potential contributing role of Rainbow Smelt as a putative predator on larval Cisco in Lakes Michigan and Huron.

## Acknowledgments

We thank Sofia Dabrowski, Tara Bell, Scott Nelson, and Wendylee Stott for retrieving historical datasheets and documents. Daniel Yule provided comments for an earlier version of this

manuscript. Any use of trade, firm, or product names is for descriptive purposes only and does not imply endorsement by the U.S. Government.

## Author Contributions

**Conceptualization:** Yu-Chun Kao, David B. Bunnell, Owen T. Gorman, Randy L. Eshenroder.

**Data curation:** Yu-Chun Kao, Renee E. Renauer-Bova, David B. Bunnell, Owen T. Gorman.

**Formal analysis:** Yu-Chun Kao, Renee E. Renauer-Bova.

**Funding acquisition:** Yu-Chun Kao, David B. Bunnell, Owen T. Gorman.

**Investigation:** Yu-Chun Kao, Renee E. Renauer-Bova, David B. Bunnell, Owen T. Gorman, Randy L. Eshenroder.

**Methodology:** Yu-Chun Kao, David B. Bunnell, Owen T. Gorman.

**Project administration:** Yu-Chun Kao, David B. Bunnell, Owen T. Gorman.

**Resources:** David B. Bunnell, Owen T. Gorman.

**Supervision:** Yu-Chun Kao, David B. Bunnell, Owen T. Gorman.

**Validation:** Yu-Chun Kao, Renee E. Renauer-Bova.

**Visualization:** Yu-Chun Kao, Renee E. Renauer-Bova.

**Writing – original draft:** Yu-Chun Kao, Renee E. Renauer-Bova, David B. Bunnell, Owen T. Gorman, Randy L. Eshenroder.

**Writing – review & editing:** Yu-Chun Kao, Renee E. Renauer-Bova, David B. Bunnell, Owen T. Gorman, Randy L. Eshenroder.

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
