## [Decision Letter · Decision Letter 0]

12 Jun 2022

PONE-D-21-35400Distributions of cisco (Coregonus artedi) in the upper Great Lakes in the mid-twentieth century, when populations were in declinePLOS ONE

Dear Dr. Kao,

Thank you for submitting your manuscript to PLOS ONE. After careful consideration, we feel that it has merit but does not fully meet PLOS ONE’s publication criteria as it currently stands. Therefore, we invite you to submit a revised version of the manuscript that addresses the points raised during the review process.

As two reviewers mentioned, the methodology should be more clearly explained. So, please carefully read the comments from the two reviewers and rewrite the ms.==============================

We look forward to receiving your revised manuscript.

Kind regards,

Syuhei Ban

Academic Editor

PLOS ONE

Journal Requirements:

2. We note that Figures 1-5 in your submission contain [map/satellite] images which may be copyrighted. All PLOS content is published under the Creative Commons Attribution License (CC BY 4.0), which means that the manuscript, images, and Supporting Information files will be freely available online, and any third party is permitted to access, download, copy, distribute, and use these materials in any way, even commercially, with proper attribution. For these reasons, we cannot publish previously copyrighted maps or satellite images created using proprietary data, such as Google software (Google Maps, Street View, and Earth). For more information, see our copyright guidelines: http://journals.plos.org/plosone/s/licenses-and-copyright.

 a. You may seek permission from the original copyright holder of Figures 1-5 to publish the content specifically under the CC BY 4.0 license. 

Reviewers' comments:

Reviewer's Responses to Questions

**Comments to the Author**

1. Is the manuscript technically sound, and do the data support the conclusions?

Reviewer #1: No

Reviewer #2: Yes

2. Has the statistical analysis been performed appropriately and rigorously? 

Reviewer #1: Yes

Reviewer #2: Yes

3. Have the authors made all data underlying the findings in their manuscript fully available?

Reviewer #1: Yes

Reviewer #2: Yes

4. Is the manuscript presented in an intelligible fashion and written in standard English?

Reviewer #1: No

Reviewer #2: Yes

5. Review Comments to the Author

Reviewer #1: General comments

This study tried to reconstruct the spatial and seasonal dynamics of cisco (Coregonus artedi) in the upper Great Lakes in the mid 20th century from the historical datasets using generalized additive models (GAMs). The statistical analyses are technically sound and the presentations of results are mostly clear. However, I have two concerns. 1) I could not fully understand why and how the authors tried to predict (reconstruct) both cisco catch and its CPUE. The conceptual and technical differences between them are clearly described. 2) The results demonstrating the spatial-temporal overlap with potential competitve species are just descriptive, so that it would be difficult to justify the authors' conclusion.

Specific comments

Line 156: catch and CPUE; what is the main difference in this paper; although I know the general definitions but specific meaning here is unclear.

Line 166: gill-net surveys?

Line 317: The authors would want to show the correlation between observed cisco catch and its CPUE if the survey period is overlaped.

Line 321: In addition to the numerical values of % explained of the deviance. The readers want to see the direct comparison between predicted and observed values, with e.g. scatter plots with 1:1 line. Or, the observed values can be added in the panels b-m in Fig.2.

Line 326: I could not find the detailed information for these adjustments in Method section.

Line 371: I am confused here. Why do the authors call this quantity as "catch" instead of CPUE, even when catch was standardized per one lift. Then it also comes back to the fundamental questions. What are the intensions to use these two datasets and to predict these two indices?

Line 407: The results shown in Figs. 4-5 are descriptive so that it is difficult to see when and where the overlap was large. The authors could calculate some indices to quantify the degree of the distribution overlap.

Reviewer #2: = Review of ``Distributions of cisco (Coregonus artedi) in the upper Great Lakes in the mid-twentieth century, when populations were in decline'' by Kao et al. (PONE-D-21-35400) =

== General comments ==

Authors analyzed historical data and estimated lake wide distribution of cisco in the upper Great Lakes. Results out of contemporary statistical methods are worthwhile without question. The objective of this study is clearly achieved. While I am not familiar with the GAM model per se, the statistical methods appears to be legitimate. On the other hand, I feel the backgrounds of the study are not well described that hampered me to better understand the results. Below are specific comments.

== Detailed comments ==

* I presume that the historical catch data analyzed in this study are Coregonus artedi only. Is is correct? Any chance to mix other cisco species in the data? Please clarify. If there are chances the data mixed other species, please state so.

* Taxonomic background of Coregonus spp. or ciscoes is better to be explained. Otherwise readers may be confused.

* line 57: I couldn't understand what a species per se is diverse probably due to the above confusion. Please clarify.

* line 216: by lake -> for each lake

* line 221-: Why these criteria were taken for the model selection? Please clarify. I am questioning this because in the previous paper by the same author group adopted AIC in stead of p values.

* line 231: ln(Catch)- I would guess there are null data in the catch. If so, ln(Catch) goes to negative infinity.

* line 301: Is this mean ``Unidentified Coregonus spp.'' was not included in the analylsys? Please clarify.

* Fig. 2: interpretation of y-axes is not clear while I understand it is adjusted by factors other than those in x-axis. For example, ln(Catch) ~ -3 in the main basin of lake Michigan is hardly grasped. I would guess it is a hypothetical value adjusted to unrealistic conditions (i.e., mesh size = 0, night = 0 ...). If so, it is better to show realistic catch value by further adjustment, or at least specify the particular adjusted conditions. The same standardization criteria as in Fig. 3 would be better in these figures.

* line 448: Are the commercial catch data from Saginaw Bay? Please specify.

* line 515: Which data particularly show the declining recruitment during 1952-1962? Please specify figures or show indicative data?

* Can you briefly describe how rainbow smelt and alewife were introduced to great lakes? Such information is useful for the understanding the results and research background.

6. PLOS authors have the option to publish the peer review history of their article (what does this mean?). If published, this will include your full peer review and any attached files.

Reviewer #1: No

Reviewer #2: No

---

## [Author Response · Author response to Decision Letter 0]

31 Aug 2022

Response to Academic Editor Syuhei Ban’s comments:

Dear Dr. Kao,

Thank you for submitting your manuscript to PLOS ONE. After careful consideration, we feel that it has merit but does not fully meet PLOS ONE’s publication criteria as it currently stands. Therefore, we invite you to submit a revised version of the manuscript that addresses the points raised during the review process.

As two reviewers mentioned, the methodology should be more clearly explained. So, please carefully read the comments from the two reviewers and rewrite the ms.

Response: Thank you and the reviewers. We followed comments from the two reviews and carried out a major revision for this manuscript, especially for the “Materials and Methods” section. We believe that our methodology is clearly explained in this revised manuscript.

Response to Reviewer #1’s comments

General comments

This study tried to reconstruct the spatial and seasonal dynamics of cisco (Coregonus artedi) in the upper Great Lakes in the mid 20th century from the historical datasets using generalized additive models (GAMs). The statistical analyses are technically sound and the presentations of results are mostly clear. However, I have two concerns. 1) I could not fully understand why and how the authors tried to predict (reconstruct) both cisco catch and its CPUE. The conceptual and technical differences between them are clearly described. 2) The results demonstrating the spatial-temporal overlap with potential competitve species are just descriptive, so that it would be difficult to justify the authors' conclusion.

Response: Thank you for these comments. They are very helpful.

For (1), our generalized linear models (GAMs) were built to model catch (i.e., count data) and we accounted for fishing effort by including them as predictors. When we used the GAMs to predict catch based on a unit of effort, these predicted values of catch should be interpreted as CPUE. We revised our “Materials and methods” to make this clarification. We also revised results reporting accordingly.

For (2), we understand that this part could be too qualitative. We revised the language, especially in the Abstract, to make it clear--our results are qualitative and more research on potential negative effects of Rainbow Smelt on Cisco is still needed.

Specific comments

Line 156: catch and CPUE; what is the main difference in this paper; although I know the general definitions but specific meaning here is unclear.

Response: The catch here is defined as number of fish caught. We made it clear in the at the beginning of “Data” subsection (line 173, in the revised manuscript). We also made it clear that we modeled Cisco gill-net catch but predicted gill-net Cisco CPUE in the paragraph starting from line 269 and define one unit of effort in line 331. We also made it clear that observed bottom-trawl Cisco CPUE was calculated directly based on data, in the paragraph starting from line 358.

Line 166: gill-net surveys?

Response: This is bottom-trawl CPUE. We too found that the way we presented in “Data” subsection can be confusing after you pointed this out. We re-wrote and condensed this subsection, with a focus to describe what we have in the source data. We described how CPUE was calculated later in the “Maps of predicted gill-net Cisco CPUE” and “Maps of observed bottom-trawl CPUE” subsections. 

Line 317: The authors would want to show the correlation between observed cisco catch and its CPUE if the survey period is overlapped.

Response: You probably have this comment because of some misunderstanding caused by our poor presentation in “Data” subsection in the original version of our manuscript. The CPUE in this study is either predicted by using models fitted gill-net data or calculated based on bottom-trawl data. As the relationships between catch and CPUE have been clearly defined in “Maps of predicted gill-net Cisco CPUE” and “Maps of observed bottom-trawl CPUE” subsections, we do not think adding these correlations will not help the manuscript so we did not do it. 

Line 321: In addition to the numerical values of % explained of the deviance. The readers want to see the direct comparison between predicted and observed values, with e.g. scatter plots with 1:1 line. Or, the observed values can be added in the panels b-m in Fig.2.

Response: We added a figure (new Fig 2) to show the predicted vs observed values.

Line 326: I could not find the detailed information for these adjustments in Method section.

Response: We revised “Modeling gill-net Cisco catch” subsection to make it clear. Specifically, we re-wrote the model equations (1-3) so that predictors accounting for fishing effort could be presented and explained right after each equation.

Line 371: I am confused here. Why do the authors call this quantity as "catch" instead of CPUE, even when catch was standardized per one lift. Then it also comes back to the fundamental questions. What are the intensions to use these two datasets and to predict these two indices?

Response: As we addressed in the response to your general comments, our generalized linear models (GAMs) were built to model catch (count data) and we accounted for fishing effort by including them as predictors. When we used the GAMs to predict catch based on a unit of effort, these predicted values of catch should be interpreted as CPUE. The manuscript has been revised accordingly.

We have CPUE derived from gill-net and bottom-trawl data because these three species have different catchability to different gears. Gill nets could catch Ciscoes well but not for Alewives and Rainbow Smelts, while Bottom-trawl could catch Rainbow Smelts and Alewives well, but not for Ciscoes. Thus, use bot data would be helpful to understand the full picture. 

Line 407: The results shown in Figs. 4-5 are descriptive so that it is difficult to see when and where the overlap was large. The authors could calculate some indices to quantify the degree of the distribution overlap.

Response: The overlap between bottom-trawl CPUE could be misleading because of the low catchability of Cisco to bottom-trawling. It would be more meaningful to investigate the overlaps between gill-net Cisco CPUE and bottom-trawl alewife CPUE and between gill-net Cisco CPUE and bottom-trawl rainbow smelt CPUE. However, quantifying the distribution overlaps based data collected from two different surveys (using different gears and at different stations) is beyond the scope of this study. Thus qualitatively described these overlaps in Results (in line 498 and line 524) and based on the maps of predicted gill-net Cisco CPUE (Fig 6) and maps of observed bottom-trawl CPUE of Alewife and Rainbow Smelt (Figs. 7 and 8).

 

Response to Reviewer #2’s comments

Reviewer #2: = Review of ``Distributions of cisco (Coregonus artedi) in the upper Great Lakes in the mid-twentieth century, when populations were in decline'' by Kao et al. (PONE-D-21-35400) =

== General comments ==

Authors analyzed historical data and estimated lake wide distribution of cisco in the upper Great Lakes. Results out of contemporary statistical methods are worthwhile without question. The objective of this study is clearly achieved. While I am not familiar with the GAM model per se, the statistical methods appears to be legitimate. On the other hand, I feel the backgrounds of the study are not well described that hampered me to better understand the results. Below are specific comments.

Response: Thanks for your comments. We followed your specific comments to provide more details on the taxonomy of coregonine fishes in the Great Lakes and the introduction and dispersal of Alewife and Rainbow smelt. We believe the readability of this manuscript has been improved.

== Detailed comments ==

* I presume that the historical catch data analyzed in this study are Coregonus artedi only. Is it correct? Any chance to mix other cisco species in the data? Please clarify. If there are chances the data mixed other species, please state so.

Response: Yes, it is correct. We only analyzed catch data for Coregonus artedi. 

The mix of Cisco with other species in our data is unlikely. Researchers carrying out these surveys were very experienced in identifying coregonine fishes, as coregonine fishes were diverse and abundant at that time. In the source data, large (> 152 mm) coregonine fishes were identified to species level while small (≤ 152 mm) ones were grouped into a category “Unidentified Coregonus”. Thus, although misidentification is still possible, it is unlikely to have Cisco and other coregonine species mixed up in a systematic manner in our data. We made this clear in Discussion (line 653) 

* Taxonomic background of Coregonus spp. or ciscoes is better to be explained. Otherwise readers may be confused.

Response: Thank you for this comment. We added a paragraph describing the taxonomic background of Coregonus spp. in the Great Lakes in Introduction (line 56).

* line 57: I couldn't understand what a species per se is diverse probably due to the above confusion. Please clarify.

Response: Diverse here means Cisco or C. artedi was diverse in morphology. We made it clear in the revised Introduction (line 83).

* line 216: by lake -> for each lake

Response: We revised the sentence accordingly (line 257).

* line 221-: Why these criteria were taken for the model selection? Please clarify. I am questioning this because in the previous paper by the same author group adopted AIC in stead of p values.

Response: We used AIC in our previous study because at that time we did not know what predictors we should include in the model. Thus, we evaluated all 12 potential models, from simple to complex, at once. In this case, using AIC is a more appropriate method as the number of predictors varied among models. 

In current study, we could start with what we already learned from the 1930-1932 study. This means that we could hypothesize that Cisco distributions in the upper Great Lakes in the study period 1952-1962 were affected by the same important predictors as those we found our previous study. In this case, a backward selection method is more appropriate as we have a lot of confidence on our full model. We made this clear in the paragraph starting from line 257.

* line 231: ln(Catch)- I would guess there are null data in the catch. If so, ln(Catch) goes to negative infinity.

Response: We assumed that “null data” here means zero catch. If “null data” means missing values, this is not an issue because there are no missing values in Catch.

If “null data” here means zero catch, the scenario “ln(Catch) goes to negative infinity” would not happen because the data were never transformed in the parameter estimation process of generalized linear or additive models (GLMs or GAMs). Data transformation is common when linear models are used. For example, ln(Catch + 1) is a reasonable way to transform our catch data. In this case, however, ln(Catch + 1) is treated as observations in the parameter estimation process. Specifically, we would not estimate the expected values of Catch but the expected values of ln(Catch + 1), which would make it difficult to back calculate the expected values Catch. The issues with log-transformation of data have been discussed widely (e.g., Duan 1983).

In GLMs and GAMs, the “link function” (e.g., the “ln(.)” we used this paper) is a function of the expected value of the response variable. Parameters of GLMs and GAMs are estimated based on maximum likelihood methods, in which the data for the response variable are not log-transformed. The ln(Catch) here in the figure means the log-transformed expected values of Catch, not the expected values of ln(Catch). The estimated values of Catch can be very close to zero but are always positive, as our GAMs are based the Tweedie distribution. This is shown in Fig 2 in the revised manuscript. 

We did not revise the manuscript in response to this comment as details about link function in GLM and GAM has been well discussed in Wood (2017).

Duan N. Smearing estimate: a nonparametric retransformation method. Journal of the American Statistical Association. 1983;78:605-10.

Wood SN. Generalized additive models: an introduction with R. Second ed. Boca Raton, Florida: CRC Press; 2017.

* line 301: Is this mean ``Unidentified Coregonus spp.'' was not included in the analyses? Please clarify.

Response: No, it was not. In fact, this should have been clarified earlier in this paper. We added more details in the “Data” subsection, starting from line 195.

* Fig. 2: interpretation of y-axes is not clear while I understand it is adjusted by factors other than those in x-axis. For example, ln(Catch) ~ -3 in the main basin of lake Michigan is hardly grasped. I would guess it is a hypothetical value adjusted to unrealistic conditions (i.e., mesh size = 0, night = 0 ...). If so, it is better to show realistic catch value by further adjustment, or at least specify the particular adjusted conditions. The same standardization criteria as in Fig. 3 would be better in these figures.

Response: Thanks for this comment. We revised Fig 2 accordingly be splitting in into three figures (now Figs 3-5). We also added estimated intercepts in the figure captions.

* line 448: Are the commercial catch data from Saginaw Bay? Please specify.

Response: Yes. We revised the sentence accordingly (line: 555).

* line 515: Which data particularly show the declining recruitment during 1952-1962? Please specify figures or show indicative data?

Response: Thanks for this question. In fact, this argument could be an over-stretch from our results, which have nothing about Cisco larvae. We revised the language and delete the discussion about Rainbow Smelt predation of Cisco larvae (line: 620). 

* Can you briefly describe how rainbow smelt and alewife were introduced to great lakes? Such information is useful for the understanding the results and research background.

Response: Sure. We briefly described the dispersals of rainbow smelt and alewife in the upper Great Lakes in Introduction (line 116).

---

## [Decision Letter · Decision Letter 1]

20 Sep 2022

PONE-D-21-35400R1Distributions of Cisco (Coregonus artedi) in the upper Great Lakes in the mid-twentieth century, when populations were in declinePLOS ONE

Dear Dr. Kao,

Thank you for submitting your manuscript to PLOS ONE. After careful consideration, we feel that it has merit but does not fully meet PLOS ONE’s publication criteria as it currently stands. Therefore, we invite you to submit a revised version of the manuscript that addresses the points raised during the review process.

The revised ms has been well improved according to reviewers' suggestions. But, reviewer #1 still concern about the methodology part. The descriptions should be clearer. Please carefully read the comments by the reviewer and rewrite the ms again.==============================

We look forward to receiving your revised manuscript.

Kind regards,

Syuhei Ban

Academic Editor

PLOS ONE

Journal Requirements:

Reviewers' comments:

Reviewer's Responses to Questions

**Comments to the Author**

1. If the authors have adequately addressed your comments raised in a previous round of review and you feel that this manuscript is now acceptable for publication, you may indicate that here to bypass the “Comments to the Author” section, enter your conflict of interest statement in the “Confidential to Editor” section, and submit your "Accept" recommendation.

Reviewer #1: (No Response)

Reviewer #2: All comments have been addressed

2. Is the manuscript technically sound, and do the data support the conclusions?

Reviewer #1: Partly

Reviewer #2: Yes

3. Has the statistical analysis been performed appropriately and rigorously? 

Reviewer #1: Yes

Reviewer #2: Yes

4. Have the authors made all data underlying the findings in their manuscript fully available?

Reviewer #1: Yes

Reviewer #2: Yes

5. Is the manuscript presented in an intelligible fashion and written in standard English?

Reviewer #1: No

Reviewer #2: Yes

6. Review Comments to the Author

Reviewer #1: Although most of authors' responses are satisfactory, I am not yet fully convinced by the response in the letter and the corresponding revision in the text, regading to the method to estimate catch and predict CPUE. The authors would need addition revision for this points to make clearer the method and ensure the reproducibility of the results.

On one hand, in the revised text (line 279-) and Eqn. 1, it reads that o(LNA) is the variable related to the fishing efforts, and the "first group of predictors" (in line 274-275) are o(LNA) and s_m(Mesh) in the eqn.1. On the other hand, when the authors mentioned the method to calculate CPUE (line 331-), the authors state that "gill-net Cisco CPUE as the predicted Cisco catchbased on predictors that represent one unit of gill-net fishing effort, which we defined as one lift of a gang of bottom-set gill nets, with a total mesh area of 100 m2, a mesh size of 51 mm, and a fishing duration of one night". I am not sure if these two descriptions are consistent or not; how did the authors exract the necessary information (=one lift of a bottom-set gill nets with the total mesh area of 100 m2, a mesh size of 51 mm, and a one night duration) from only two variables in the model (i.e., o(LNA) and s_m(Mesh))?

Becauase I am not a native English speaker, I cannot judge whether this issues come from English language or (more seriously) from logical or methodologiral flaws. The authors may want to check again this issue to give much clearer presentation and make sure the results are reproducible.

Reviewer #2: All comments and concerns to the previous version are adequately addressed. I have no reservations about the publication.

7. PLOS authors have the option to publish the peer review history of their article (what does this mean?). If published, this will include your full peer review and any attached files.

Reviewer #1: No

Reviewer #2: No

---

## [Author Response · Author response to Decision Letter 1]

24 Sep 2022

Response to Reviewer #1’s comments

Although most of authors' responses are satisfactory, I am not yet fully convinced by the response in the letter and the corresponding revision in the text, regarding to the method to estimate catch and predict CPUE. The authors would need addition revision for this points to make clearer the method and ensure the reproducibility of the results. 

On one hand, in the revised text (line 279-) and Eqn. 1, it reads that o(LNA) is the variable related to the fishing efforts, and the "first group of predictors" (in line 274-275) are o(LNA) and s_m(Mesh) in the eqn.1. On the other hand, when the authors mentioned the method to calculate CPUE (line 331-), the authors state that "gill-net Cisco CPUE as the predicted Cisco catch based on predictors that represent one unit of gill-net fishing effort, which we defined as one lift of a gang of bottom-set gill nets, with a total mesh area of 100 m2, a mesh size of 51 mm, and a fishing duration of one night". I am not sure if these two descriptions are consistent or not; how did the authors exract the necessary information (=one lift of a bottom-set gill nets with the total mesh area of 100 m2, a mesh size of 51 mm, and a one night duration) from only two variables in the model (i.e., o(LNA) and s_m(Mesh))?

Because I am not a native English speaker, I cannot judge whether these issues come from English language or (more seriously) from logical or methodological flaws. The authors may want to check again this issue to give much clearer presentation and make sure the results are reproducible.

Response: We added more explanation in the Materials and methods (starting from line 295 and starting from line 339) to clarify how CPUE was estimated. Generally, each of the Lake Superior, Lake Michigan, and Saginaw Bay full models includes different predictors for fishing effort adjustment. Thus, different inputs associated with fishing effort could be required for each selected model to estimate gill-net Cisco CPUE. For example, Lake Superior full model only included o(LNA) and sm(Mesh) as predictors for fishing effort adjustment because only data collected from gill nets set on the bottom and then lifted after one night were included in the model fitting. Thus, Catch in the Lake Superior model already represents number of Cisco caught from one lift of a gang of bottom-set gill nets, with a fishing duration of one night. Then, if the full Lake Superior model was selected, only a total gill-net mesh area of 100 m2 and a mesh size of 51 mm were required as inputs for o(LNA) and sm(Mesh), respectively, to estimate gill-net Cisco CPUE.

---

## [Decision Letter · Decision Letter 2]

30 Sep 2022

Distributions of Cisco (Coregonus artedi) in the upper Great Lakes in the mid-twentieth century, when populations were in decline

PONE-D-21-35400R2

Dear Dr. Kao,

We’re pleased to inform you that your manuscript has been judged scientifically suitable for publication and will be formally accepted for publication once it meets all outstanding technical requirements.

Kind regards,

Syuhei Ban

Academic Editor

PLOS ONE

Additional Editor Comments (optional):

Reviewers' comments:

Reviewer's Responses to Questions

**Comments to the Author**

1. If the authors have adequately addressed your comments raised in a previous round of review and you feel that this manuscript is now acceptable for publication, you may indicate that here to bypass the “Comments to the Author” section, enter your conflict of interest statement in the “Confidential to Editor” section, and submit your "Accept" recommendation.

Reviewer #1: All comments have been addressed

2. Is the manuscript technically sound, and do the data support the conclusions?

Reviewer #1: Yes

3. Has the statistical analysis been performed appropriately and rigorously? 

Reviewer #1: Yes

4. Have the authors made all data underlying the findings in their manuscript fully available?

Reviewer #1: Yes

5. Is the manuscript presented in an intelligible fashion and written in standard English?

Reviewer #1: Yes

6. Review Comments to the Author

Reviewer #1: Thank you very much for your revision. The revision to make clearer the method of estimating CPUE is now super clear.

7. PLOS authors have the option to publish the peer review history of their article (what does this mean?). If published, this will include your full peer review and any attached files.

Reviewer #1: No

---

## [Editor Report · Acceptance letter]

13 Dec 2022

PONE-D-21-35400R2 

Distributions of Cisco *(Coregonus artedi)* in the upper Great Lakes in the mid-twentieth century, when populations were in decline 

Dear Dr. Kao:

I'm pleased to inform you that your manuscript has been deemed suitable for publication in PLOS ONE. Congratulations! Your manuscript is now with our production department. 

Kind regards, 

on behalf of

Dr Syuhei Ban 

Academic Editor

PLOS ONE